# Body size as a metric for the affordable world

**Xinran Feng**[1†], **Shan Xu**[2†], **Yuannan Li**[1], **Jia Liu**[1]*

[1]Department of Psychology & Tsinghua Laboratory of Brain and Intelligence, Tsinghua University, Beijing, China; [2]Faculty of Psychology, Beijing Normal University, Beijing, China

**Abstract** The physical body of an organism serves as a vital interface for interactions with its environment. Here, we investigated the impact of human body size on the perception of action possibilities (affordances) offered by the environment. We found that the body size delineated a distinct boundary on affordances, dividing objects of continuous real-world sizes into two discrete categories with each affording distinct action sets. Additionally, the boundary shifted with imagined body sizes, suggesting a causal link between body size and affordance perception. Intriguingly, ChatGPT, a large language model lacking physical embodiment, exhibited a modest yet comparable affordance boundary at the scale of human body size, suggesting the boundary is not exclusively derived from organism-environment interactions. A subsequent fMRI experiment offered preliminary evidence of affordance processing exclusively for objects within the body size range, but not for those beyond. This suggests that only objects capable of being manipulated are the objects capable of offering affordance in the eyes of an organism. In summary, our study suggests a novel definition of object-ness in an affordance-based context, advocating the concept of embodied cognition in understanding the emergence of intelligence constrained by an organism's physical attributes.

*For correspondence:
liujiathu@tsinghua.edu.cn

†These authors contributed equally to this work

Competing interest: The authors declare that no competing interests exist.

## eLife assessment

This paper presents **valuable** findings that shed light on the mental organisation of knowledge about real-world objects. It provides diverse, if **incomplete** and tentative, evidence from behaviour, brain, and large language models that this knowledge is divided categorically between relatively small objects (closer to the relevant scale for direct manipulation) and larger objects (further from the typical scope of human affordances for action).

## Introduction

*Man is the measure of all things. - Protagoras*

The assertion by the ancient Greek philosopher Protagoras highlights the notion that reality is defined by how the world is perceived by humans. A contemporary interpretation of this statement is the embodied theory of cognition (e.g. *Chemero, 2013*; *Gallagher, 2017*; *Yu et al., 2005*; *Wilson, 2002*; *Varela et al., 2017*), which, diverging from the belief that size and shape are inherent object features (e.g. *Op de Beeck et al., 2008*; *Konkle and Oliva, 2011*), posits that human body scale (e.g. size) constrains the perception of objects and the generation of motor responses. For instance, humans evaluate the climbability of steps based on their leg length (*Mark, 1987*; *Warren, 1984*), and determine the navigability of apertures according to the critical aperture-to-shoulder-width ratio (*Warren and Whang, 1987*). Additionally, grasping strategies have been shown to be contingent upon object size relative to one's body (*Cesari and Newell, 2000*; *Newell et al., 1989*) or hand size (*Castiello et al., 1993*; *Tucker and Ellis, 2004*). However, the question of how object perception is

influenced by the relative size of objects in relation to the human body remains open. Specifically, it is unclear whether this relative size simply acts as a continuous variable for locomotion reference, or if it affects differentiating and organizing object representation based on their ensued affordances.

To underscore the latter point, *Gibson, 1979*, the pioneer of embodied cognition research, stated that "Detached objects must be comparable in size to the animal under consideration if they are to afford behavior (p.124)." This implies that an object's affordance, encompassing all action possibilities offered to an animal, is determined by the object's size relative to the animal's size rather than its real-world size. For instance, in a naturalistic environment, such as a picnic scene shown in *Figure 1a*, there may exist a qualitative distinction between objects within (the objects with warm tints in *Figure 1a*) and beyond (those with cold tints) the size range of humans. Only objects within the range, such as the apple, the umbrella, and the bottles, may afford actions, while those beyond this range, such as the trees and the tent, are largely viewed as part of the environment. Consequently, visual perception may be ecologically constrained, and the body may serve as a metric that facilitates meaningful engagement with the environment by differentiating objects that are accessible for interactions from those not. Further, grounded cognition theory (see *Barsalou, 2008* for a review) suggests that the outputs of such differentiation might transcend sensorimotor processes and integrate into supramodal concepts and language. From this perspective, we proposed two hypotheses: first, the affordance of objects will exhibit a qualitative difference between objects within and beyond the size range of an organism's body; second, affordance-related neural activity will emerge exclusively for objects within the organism's size range.

To test these hypotheses, we first measured the affordance of a diverse array of objects varying in real-world sizes (e.g. *Konkle and Oliva, 2011*). We found a dramatic decline in affordance similarity between objects within and beyond the human body size range, as these objects afforded distinct sets of action possibilities. Notably, the affordance boundary varied in response to the imagined body sizes and showed supramodality. It could also be attained solely through language, as evidenced by the large language model (LLM), ChatGPT (*OpenAI, 2023*). A subsequent fMRI experiment corroborated the qualitative difference in affordances demarcated by the body size, as affordances of objects within humans' size range, but not those beyond, were represented in both dorsal and ventral visual streams of the brain. This study advances our understanding of the role of body size in shaping object representation and underscores the significance of body size as a metric for determining object affordances that facilitates meaningful engagement with the environment.

## Results

To illustrate how human body size affects object affordances with different sizes, we first characterized the affordances of a set of daily objects. In each trial, we presented a matrix consisting of nine objects and asked participants to report which objects afforded a specific action (e.g. sit-able: a chair, a bed, a skateboard, but not a phone, a laptop, an umbrella, a kettle, a plate, or a hammer; *Figure 1b*). In this task, there were 14 actions commonly executed in daily life and 24 object images from the THINGS database (*Hebart et al., 2019*), with sizes ranging from size rank 2 to 8 according to *Konkle and Oliva, 2011*'s classification. These objects covered real-world sizes from much smaller (17 cm on average, rank 2) to orders of magnitude larger (5317 cm on average, rank 8) than the human body size (see Materials and methods for details). Consequently, affordances for each object were indexed by a 14-dimensional action vector, with the value for each dimension representing the percentage of participants who agreed on a certain action being afforded by the object (e.g. 88% for the action of grasping on a hammer indicating 88% of participants agreed that a hammer affords grasping). *Figure 1—figure supplement 1* shows the affordances of two example objects.

An affordance similarity matrix was then constructed where each cell corresponded to the similarity in affordances between a pair of objects (*Figure 1c*). A clustering analysis revealed a two-cluster structure. Visual inspection suggested that the upper-left cluster consisted of objects smaller than human body size (red labels), and the lower-right cluster contained objects larger than human body size (green labels). Critically, the between-cluster similarity in the affordance similarity matrix approached zero, suggesting a division in affordances located near the body size. To quantify this observation, we calculated the similarity in affordances between each neighboring size rank. Indeed, we identified a clear trough in affordance similarity, dropping to around zero, between size rank 4 (77cm on average) and 5 (146cm on average), which was significantly smaller than that between size rank 3 and

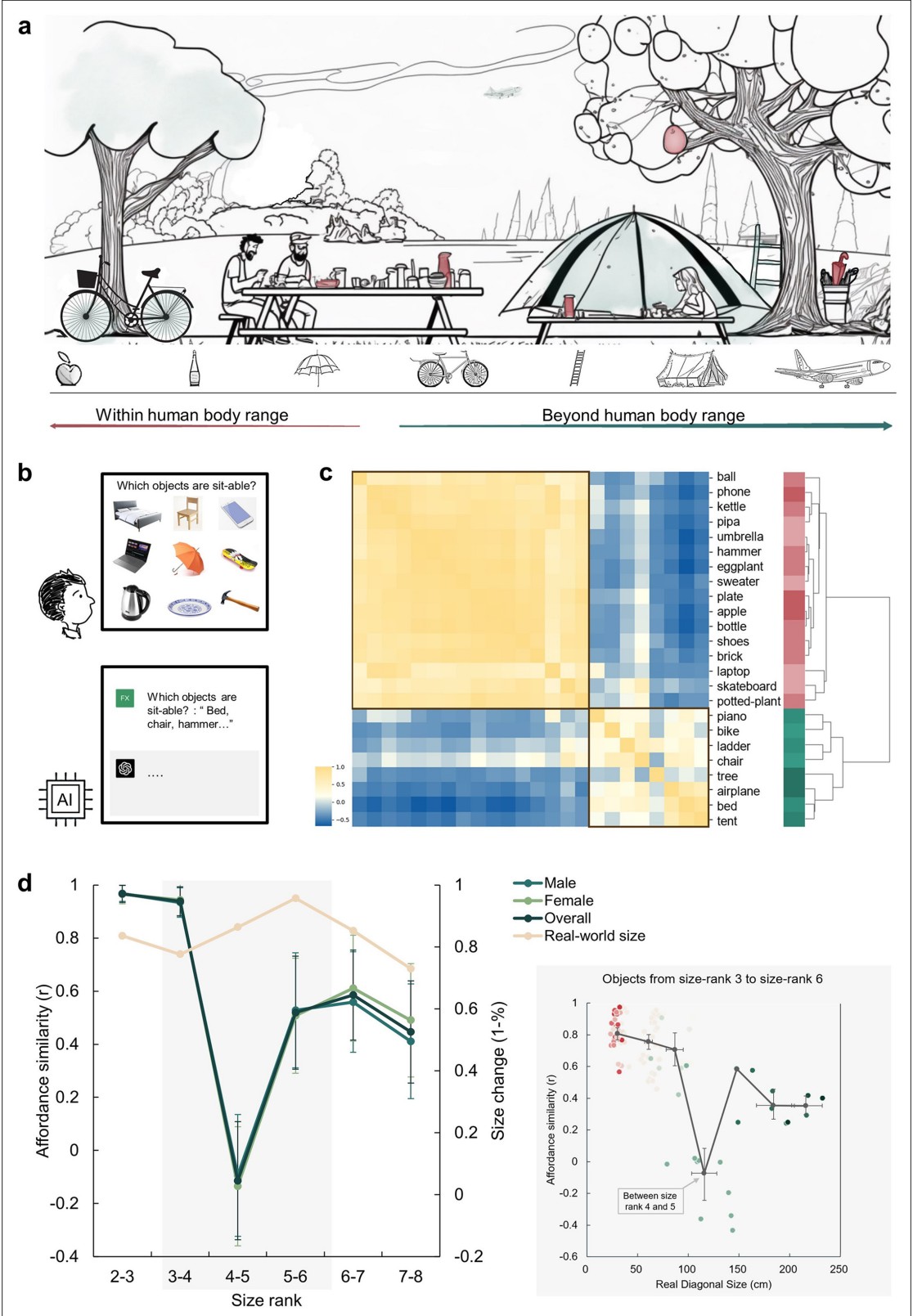

**Figure 1.** An affordance boundary in the affordable world. (**a**) An illustration of a picnic scene, featuring objects of various sizes relative to human body. Example objects within the normal body size range are painted red, and those beyond green. We hypothesized qualitative differences between perceived affordances of these two kinds of objects. (**b**) A demonstration of the object-action relation judgement task for human participants (top) and AI models (bottom). The question in the task for human participants was presented in Chinese. (**c**) The representational similarity matrix (RSM) for

*Figure 1 continued*

objects based on human rating of affordance similarity. Object sizes are denoted with red to green. Two primary clusters emerged in the clustering analysis of the similarity pattern are outlined with black boxes. (**d**) *Left panel*: The overall affordance similarity and that of each gender (left y-axis) as well as real-world size similarity (right y-axis) between neighboring size ranks. The error bars represent the standard error (SE), computed from sample size $n = 14$. *Right panel*: The point clouds of pairwise correlations between objects from the same rank or neighboring ranks. Each colored dot represents the affordance similarity (y-axis) and the average real-world size (x-axis) of a specific object pair. The grey dots indicate the averaged size (x-axis) and pairwise similarity (y-axis) of object pairs in different rank compositions. Left to right: both from size rank 3, from size rank 3 and 4, both from size rank 4, from size rank 4 and 5, both from size rank 5, from size rank 5 and 6, and both from size rank 6. The horizontal error bars represent 95% confidence interval (CI) of the averaged object size in each pair, and the vertical error bars denote the CI of pairwise affordance similarity (sample size $n = 100$).

The online version of this article includes the following figure supplement(s) for figure 1:

**Figure supplement 1.** Two exemplar objects with different affordance profiles.

4 ($Z=3.91$, $p<0.001$) and that between size rank 5 and 6 ($Z=1.66$, $p=0.048$). This trough suggested an affordance boundary between size rank 4 and 5, while affordance similarities between neighboring ranks remained high ($rs >0.45$) and did not significantly differ from each other ($ps >0.05$, all $BF_{10} < 10$) on either side of the boundary (*Figure 1d*, left panel, green lines). This pattern was evident for both genders, indicating no gender difference. Note that the abrupt change in affordance similarity across the boundary cannot be explained by changes in objects' real-world size, as the similarity in objects' real-world size was relatively stable across ranks, without any trend of a trough-shape curve (*Figure 1d*, left panel, yellow line). Intriguingly, rank 4 and rank 5 correspond to 80 cm to 150 cm, a boundary situated between these two ranks is within the range of the body size of a typical human adult. This finding suggested that objects were classified into two categories based on their affordances, with the boundary aligning with human body size.

To better locate the boundary, we focused on the affordance similarity between individual objects within size rank 3–6 (approximately ranging from 30cm to 220 cm in real-world size, the area with grey shade in *Figure 1d*), where the trough-shape curve was identified. Specifically, we traversed all pairs of objects with similar real-world diagonal sizes (from either the same rank or from neighboring ranks), calculated their average real-world size as an index of the approximate location of the boundary between this pair of objects, and plotted the affordance similarity against the average real-world size of each object pair. As shown in the inset (grey box) of *Figure 1d*, consistent with the rank-wise analysis, the abrupt decrease in affordance similarity exclusively happened between objects from size rank 4 and 5 (light green dots). The averaged real-world size in these object pairs was 104 cm (95% CI, 105–130 cm) and the affordance similarity in such object pairs was around zero. This result further narrowed the location estimation of the boundary, and demonstrated that the affordance boundary persisted at the level of individual objects.

One may argue that the location of the affordance boundary coincidentally fell within the range of human body size, rather than being directly influenced by it. To rule out this possibility, we directly manipulated participants' body schema, referring to an experiential and dynamic functioning of the living body within its environment (*Merleau-Ponty and Smith, 1962*). This experimental approach was able to establish a causal link between body size and affordance boundary, as other potential factors remained constant. Specifically, we instructed a new group of participants to imagine themselves as small as a cat (typical diagonal size: 77 cm, size rank 4, referred to as the 'cat condition'), and another new group to envision themselves as large as an elephant (typical diagonal size: 577 cm, size rank 7, referred to as the 'elephant condition') throughout the task (*Figure 2a*). A between-subject design was adopted to minimize contamination between conditions. This manipulation was effective, as evidenced by the participants' reported imagined heights in the cat condition being 42 cm (SD=25.6) and 450 cm (SD=426.8) in the elephant condition on average, respectively, when debriefed at the end of the task.

With exactly the same set of objects, a distinct shift in the affordance boundary was observed for each condition (*Figure 2b*). In the cat condition, the affordance boundary was identified between size rank 3 and 4, with affordance similarity between size rank 3 and 4 being significantly lower than that between size rank 2 and 3 ($Z=1.76$, $p=0.039$) and that between size rank 4 and 5 ($Z=1.68$, $p=0.047$). In contrast, in the elephant condition, the affordance boundary shifted to the right, as demonstrated by a decrease in affordance similarity between size rank 6 and 7, and that between size rank 7 and 8 as compared to that between size rank 5 and 6, with a trend towards significance (with size rank 6–7:

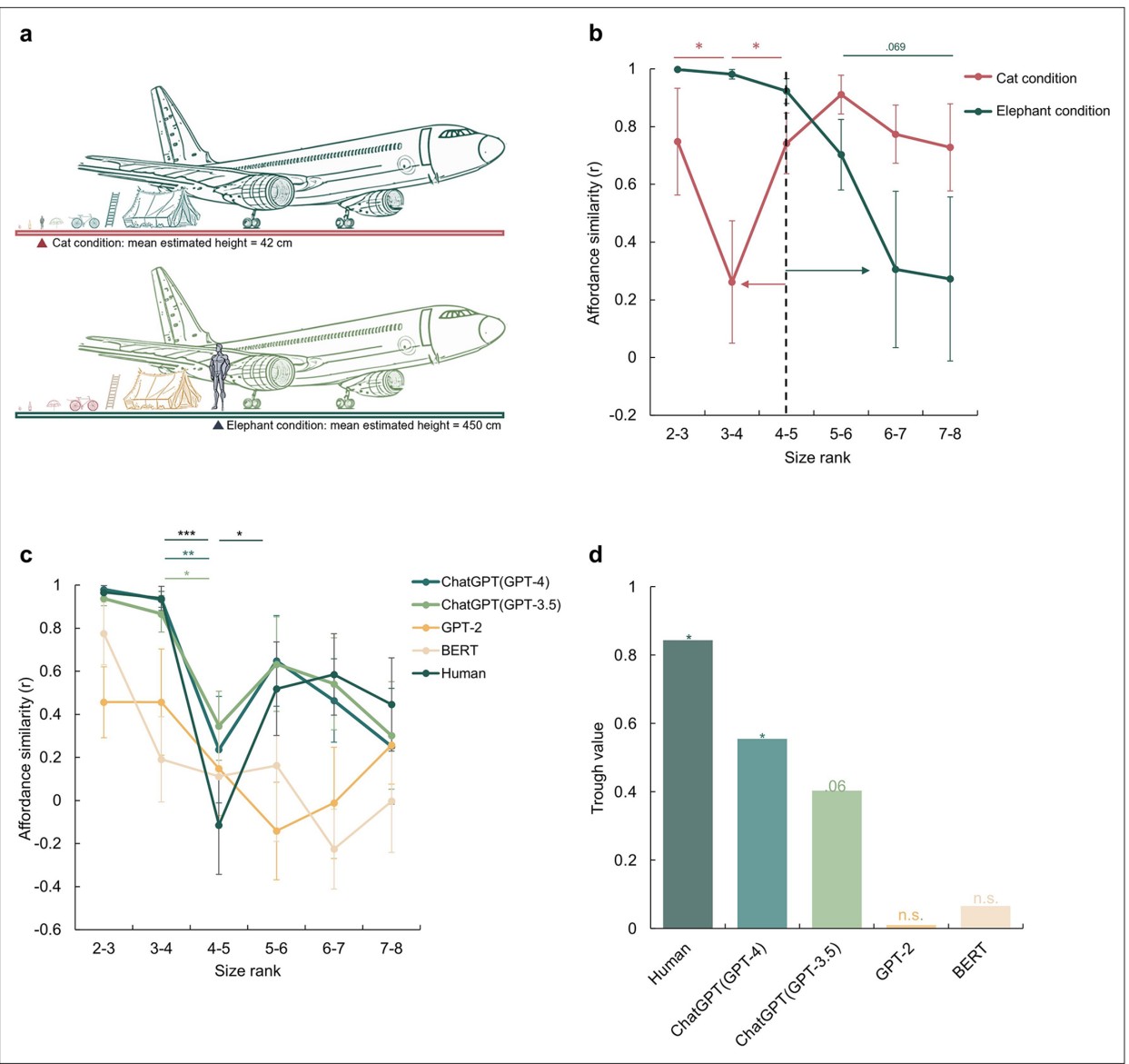

**Figure 2.** A disembodied origin of the affordance boundary. (**a**) The schematic diagram of the imagined size in the cat condition (top) and the elephant condition (bottom), with the mean estimated height reported by participants for each condition. (**b**), The affordance similarity between neighboring size ranks for manipulated body sizes (Red line: cat-size body; Green line: elephant-size body). The dashed line marks the boundary of the human-size body. The red and green arrows indicate the corresponding boundary shift in each condition. (**c**), The affordance similarity between neighboring size ranks for different large language models, and human data from *Figure 1d* was re-drawn as a reference. The stars indicate significant contrasts between affordance similarities between neighboring data points. (**d**), The trough value of each model at between size rank 4–5. The stars here indicate the significant trough value compared to zero. The error bars represent the estimated standard error (SE), computed from sample size $n = 14$. *$p<0.05$, **$p<0.01$, ***$p<0.001$.

Z=1.28, p=0.099; with size rank 7–8: Z=1.48, p=0.069). The observation that the affordance boundary shifted to the left under the cat condition and to the right under the elephant condition suggests that affordance perception is influenced even by imagined body size. Furthermore, the cognitive penetrability (*Pylyshyn, 1999*) of affordance perception implies potential susceptibility of affordance perception to semantic or conceptual transformation or modification.

To test the further speculation that the affordance boundary can be derived solely from conceptual knowledge without direct sensorimotor experience, we employed a disembodied agent, the large language model (LLM) ChatGPT (Chat Generative Pre-trained Transformer; https://openai.com/blog/chatgpt/). This model was trained on a massive corpus of language materials originated from humans, yet it can not receive any sensorimotor information from the environment. Here we asked whether

language alone would be sufficient to form an affordance boundary in ChatGPT models as well as in smaller LLMs, BERT (*Devlin et al., 2018*), and GPT-2 (*Radford et al., 2018*).

The experimental procedure was similar to that conducted with human participants, except that images were replaced by the corresponding words (see Methods). Given the randomness embedded in response generation, each model was tested 20 times to simulate the sampling of human participants. We found that the affordance similarity curves demonstrated by the ChatGPT models were both trough-shaped between size rank 4 and 5, the same location where the boundary emerged in human participants (*Figure 2c*, green lines). Further statistical analyses showed a significant difference in affordance similarity between size rank 3 and 4, and that between size rank 4 and 5 (ChatGPT (GPT-3.5): Z=1.98, p=0.024; ChatGPT (GPT-4): Z=2.73, p=0.003). The affordance similarity between size rank 4 and 5 was also lower than that between size rank 5 and 6, yet the difference did not reach the significance (ChatGPT (GPT-3.5): Z=0.96, p=0.17; ChatGPT (GPT-4): Z=1.27, p=0.10). In contrast, no trough-shaped boundary was observed in either BERT or GPT-2 (*Figure 2c*, yellow lines), despite an apparent but non-significant decrease in affordance similarity in GPT-2 between size rank 5 and 6 (*p*s >0.20). To further quantify the magnitude of the decrease in affordance similarity between the size rank 4 and 5, we measured the decrease by subtracting the similarity value at the trough from the neighboring similarity values and then subjected it to a permutation test (see Materials and methods). We found a significant decrease in affordance similarity in humans (permutation N=5000, $p\left(T > T_{obs}\right)$ = 0.015) and ChatGPT (GPT-4) (permutation N=5000, $p\left(T > T_{obs}\right)$ = 0.046), a marginal significant decrease in ChatGPT (GPT-3.5) (permutation N=5000, $p\left(T > T_{obs}\right)$ = 0.061), and no significance in either BERT or GPT-2 (*p*s >0.46, *Figure 2d*). Thus, the affordance boundary can be derived from language solely without sensorimotor information from environment. Interestingly, it appears to spontaneously emerge when the language processing ability of the LLMs surpasses a certain threshold (i.e. GPT-2 /BERT < ChatGPT models).

A further analysis on the affordances separated by the boundary revealed that objects within human body size range were primarily subjected to hand-related actions such as grasping, holding, and throwing. These affordances typically involve object manipulation with humans' effectors. In contrast, objects beyond the size range of human body predominantly afforded actions such as sitting and standing, which typically require locomotion or posture change of the whole body around or within the objects. The distinct categories of reported affordances demarcated by the boundary imply that the objects on either side of the boundary may be represented differently in the brain. We thus speculated that the observed behavioral discontinuity is likely underpinned by distinct neural activities, which give rise to these discrete 'representations' separated by the boundary.

To test this speculation, we ran an fMRI experiment with a small number of participants to preliminarily investigate the neural basis of the affordance boundary in the brain by measuring neural activity in the dorsal and ventral visual streams when participants were instructed to evaluate whether an action was affordable by an object (*Figure 3a*). Four objects were chosen from the behavioral experiment: two within the body size range (i.e. bottle and football, WITHIN condition) and the two beyond (i.e. bed and piano, BEYOND condition). Accordingly, four representative actions (to grasp, to kick, to sit, and to lift) were selected in relation to the respective objects. During the scan, the participants were asked to decide whether a probe action was affordable (e.g. grasp-able – bottle, Congruent condition) or not (e.g. sit-able – bottle, Incongruent condition) by each subsequently presented object. The congruency effect, derived from the contrast of Congruent versus Incongruent conditions, is a well-established measure of affordance processing (e.g. *Kourtis et al., 2018*).

We examined the congruency effect in two object-selective regions defined by the contrast of objects against baseline (see Materials amd methods), each representing a corresponding visual stream: the posterior fusiform (pFs) in the ventral stream, which is involved in object recognition (e.g. *Grill-Spector et al., 2000*; *Malach et al., 1995*) and objects' real-world size processing (*Konkle and Oliva, 2012*; *Snow et al., 2011*), and the superior parietal lobule (SPL) in the dorsal stream, one of the core tool network regions (e.g. *Filimon et al., 2007*; *Matić et al., 2020*). For the rest object-selective regions identified in this experiment, see *Figure 3—figure supplement 1* and *Supplementary file 1a*. A repeated-measures ANOVA with object type (WITHIN versus BEYOND) and congruency (Congruent versus Incongruent) as within-subject factors was performed for each ROI, respectively. A significant interaction between object type and congruency was observed in both ROIs (SPL: $F(1,11)$ = 15.47, p=0.002, $\eta^2$ = .58; pFs: $F(1,11)$ = 24.93, p<0.001, $\eta^2$ = .69), suggesting that these regions represented

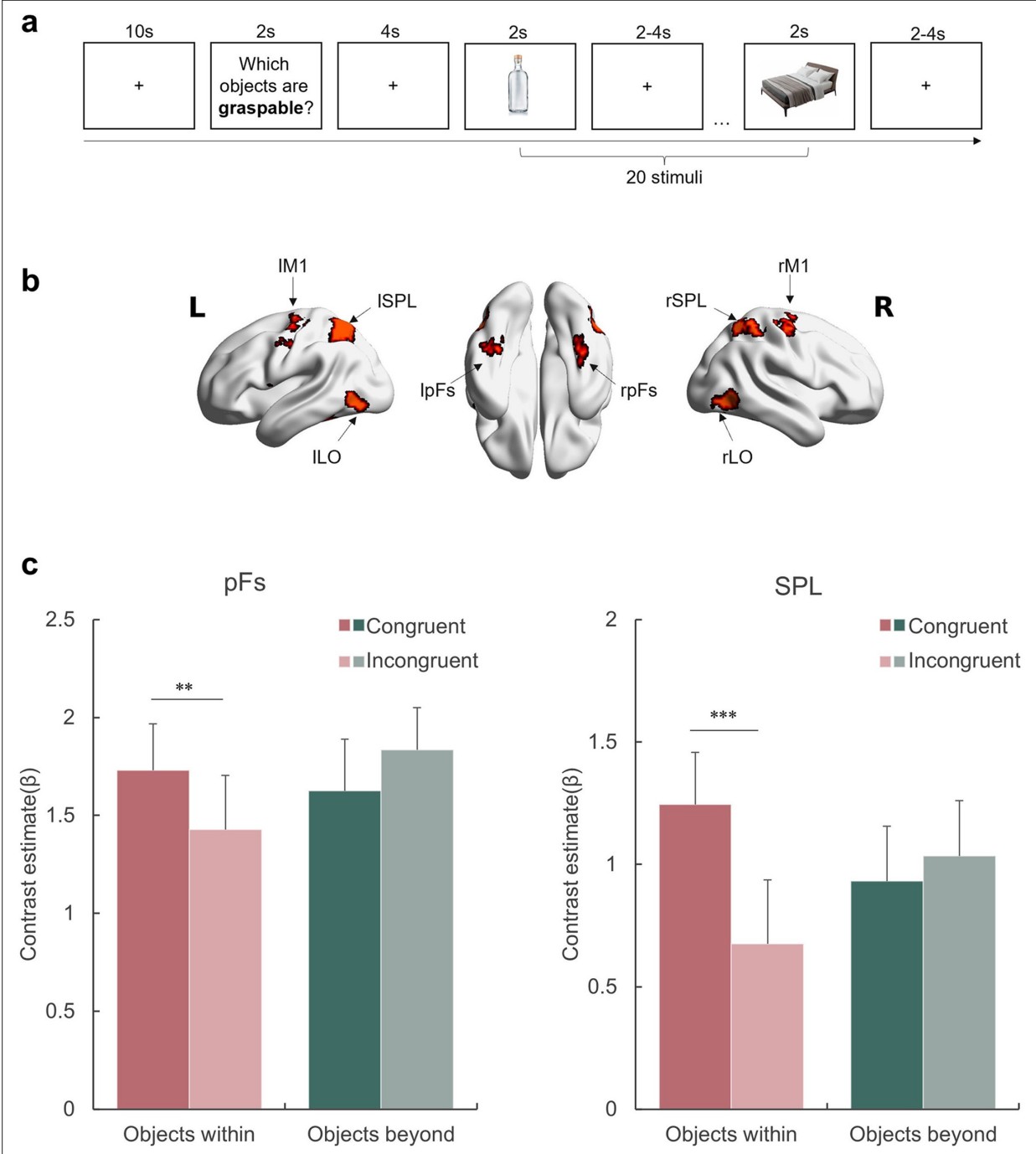

**Figure 3.** Affordance representation in the visual streams. (**a**) An example block with the probe action 'graspable'. The participants indicated whether each of the subsequently presented objects was graspable by pressing the corresponding button. The action probing question was presented in Chinese during the experiment. (**b**) The ROIs included in this experiment. (**c**) The activation of each condition in the pFs and SPL. The bars represent the contrast estimates of each condition versus baseline. The stars indicate the significant difference between congruent and incongruent conditions. *p<0.05, **p<0.01, ***p<0.001, otherwise non-significance. Error bars represent the standard error (SE), computed from sample size *n* = 12.

The online version of this article includes the following figure supplement(s) for figure 3:

**Figure supplement 1.** Brain areas showing significantly greater neural activation for objects within body size and beyond body size versus baseline.

**Figure supplement 2.** The activation in LO and M1 in response to objects within body size in the congruent and incongruent conditions, respectively.

**Figure supplement 3.** Significant brain activations of different contrasts in the whole-brain level analysis.

affordances differentially based on object type (*Figure 3c*). A post hoc simple effect analysis revealed the congruency effect solely for objects within body size range (SPL: p<0.001; pFs: p=0.021), not for objects beyond (*ps* >0.41). In addition, the main effect of object type was not significant in either ROI (*ps* >0.17), suggesting that the absence of the congruency effect for objects beyond the body size cannot be attributable to compromised engagement in viewing these objects. In addition, a whole-brain analysis was performed, and no region showed the congruency effect for the objects beyond the body size. Taken together, the affordance boundary not only separated the objects into two categories based on their relative size to human body, but also delineated the range of objects that evoked neural representations associated with affordance processing.

In addition to the pFs and SPL, we also examined the congruency effect in the lateral occipital cortex (LO), which is involved in object representation (e.g. *Grill-Spector et al., 2000*; *Konkle and Caramazza, 2013*) and provides inputs to both the pFs and SPL (*Hebart et al., 2018*). Meanwhile, the primary motor cortex (M1), which receives inputs from the dorsal stream (*Vainio and Ellis, 2020*), is involved in affordance processing (e.g. *McDannald et al., 2018*) and action executions (*Binkofski et al., 2002*). Although both the LO and M1 showed a significantly higher response to objects than baseline, no congruency effect in affordance for objects within the body size was observed (main effect of congruency: $F(1,11) = 1.74$, p=0.214, $\eta^2 = 0.13$, *Figure 3—figure supplement 2*). Therefore, it is unlikely that the representation of affordance is exclusively dictated by visual inputs or automatically engaged in motor execution. This finding suggests that affordance perception likely requires perceptual processing and is not necessarily reflected in motor execution, diverging from Gibsonian concept of direct perception.

## Discussion

One long-lasting debate on affordance centers on the distinction between representational and direct perception of affordance. An outstanding theme shared by many embodied theories of cognition is the replacement hypothesis (e.g. *van Gelder, 1998*), which challenges the necessity of representation as posited by computationalism's cognitive theories (e.g. *Fodor, 1975*). This hypothesis suggests that input is discretized/categorized and subjected to abstraction or symbolization, creating discrete stand-ins for the input (e.g. representations/states). Such representationalization would lead to a categorization between the affordable (the objects) and those beyond affordance (the environment), in contrast to the perspective offered by embodied theories. The present study probed this 'representationalization' of affordance by examining whether affordance perception introduces discontinuity and qualitative dissociation in response to continuous action-related physical features (such as object size relative to the agents), which allows sensorimotor input to be assigned into discrete states/kinds, in line with the representation-based view under the constraints of body size. Alternatively, it assessed whether activity directly mirrors the input, free from discretization/categorization/abstraction, in line with the representation-free view.

First, our study found evidence demonstrating discretization in affordance perception. Then, through the body imagination experiment, we provided causal evidence suggesting that this discretization originates from sensorimotor interactions with objects rather than amodal sources, such as abstract object concepts independent of agent motor capability. Finally, we demonstrated the supramodality of this embodied discontinuity by leveraging the recent advances in AI. We showed that the discretization in affordance perception is supramodally accessible to disembodied agents such as large language models (LLMs), which lack sensorimotor input but can access linguistic materials built upon discretized representations. These results collectively suggest that sensorimotor input undergoes discretization, as implied in the computationalism's idea of representation. Note that, these results are not contradictory to the claim of the embodied theories, as these representations do shape processes beyond the sensorimotor domain but after discretization.

This observed boundary in affordance perception extends the understanding of the discontinuity in perception in response to the continuity of physical inputs (*Harnad, 1987*; *Young et al., 1997*). Perceptual boundary has been demonstrated in various domains, such as color perception (*Bornstein and Korda, 1984*), speech-sounds perception (*Liberman et al., 1957*), and facial gender discrimination (*Campanella et al., 2001*). The boundaries reflect a fundamental adaptation of perception to facilitate categorizations necessary for an organism (*Goldstone and Hendrickson, 2010*). Our study,

for the first time, unveiled a boundary in object affordance, wherein affordance similarity across the boundary was significantly lower than that within the boundary. Critically, the boundary separating object affordances along a size axis coincided with human body size, suggesting that object affordances are characterized in a dimension scaled by human body size.

What is the function of the affordance boundary? About four decades ago, *Gibson, 1979* postulated that only objects of sizes comparable to an animal's body size are amenable to interaction and capable of providing affordances to the animal, thereby possessing ecological values that distinctly differ from those of larger objects. In this study, we expand upon this notion by arguing that the affordance boundary serves to delineate (manipulable) objects from their surrounding environment. In other words, objects within the range of an animal's body size are indeed objects in the animal's eye and possess affordances as defined by Gibson. In contrast, objects larger than that range typically surpass the animal's motor capabilities, rendering them too cumbersome for effective manipulation. Consequently, these larger objects are less likely to be considered as typical targets for manipulation by the animal, as opposed to the smaller objects. That is, they are perceived not as *the* "objects" in the animal's eye, but as part of the background environment, due to their impracticality for direct interactions. Future studies should incorporate a broader range of objects and a more comprehensive set of affordances for finer delineation of the representational discontinuity between objects and the environment.

This speculation aligns with previous fMRI studies where large objects activated the medial portion of the ventral temporal cortex (*Huang et al., 2022*; *Magri et al., 2021*), overlapping with the parahippocampus gyrus involved in scene representation (*Park et al., 2011*; *Troiani et al., 2014*), and smaller objects activated the lateral portion, such as the pFs, where the congruency effect of affordance was identified in our study. Furthermore, we found that the congruency effect was only evident for objects within the body size range, but not for objects beyond, supporting the idea that affordance is typically represented only for objects within the body size range. While it is acknowledged that the sample size of the fMRI study was small (12 participants), necessitating cautious interpretation of its results, the observed neural-level affordance discontinuity is notable. That is, qualitative differences in neural activity between objects within the affordance boundary and those beyond replicated our behavioral findings. This convergent evidence reinforced our claim that objects were discretized into two broad categories along the continuous size axis, with affordance only being manifested for objects within the boundary.

In this context, an animal's body size and sensorimotor capacity determine the boundary of manipulation, and thus, the boundary between manipulable objects and the environment. Therefore, our study provides a novel perspective on a long-standing question in psychology, cognitive science, and philosophy: what constitutes an object? Existing psychological studies, especially in the field of vision, define objects in a disembodied manner, primarily relying on their physical properties such as shape (e.g. *Op de Beeck et al., 2008*) and absolute size (e.g. *Konkle and Oliva, 2011*). Our identification of the affordance boundary presents a new source of *object-ness*: the capability of being a source of affordance under the constraints of an animal's sensorimotor capacity, which resonates the embodied influence on the formation of abstract concepts (e.g. *Barsalou, 1999*; *Lakoff and Johnson, 1980*) of objects and environment. Consistently, our fMRI data did not show the congruency effect for objects beyond the body size range, distinct from objects within this range, suggesting a categorization influenced by objects' relative size to the human body. In this respect, man is indeed the measure of all things.

The metric provided by the body size, however, was changeable when the body schema was intentionally altered through participants' imagination of possessing either a cat- or elephant-sized body, with which the participants had no prior sensorimotor experience. Importantly, they perceived new affordances in a manner as if they had had embodied experience with this new body schema. Therefore, this finding suggests that the affordance boundary is cognitively penetrable, arguing against the directness of affordance perception (e.g. *Gibson, 1979*; *Greeno, 1994*; *Prindle et al., 1980*) or the exclusive sensorimotor origin of affordances (e.g. *Gallagher, 2017*; *Thompson, 2010*; *Hutto and Myin, 2012*; *Chemero, 2013*). Further, this finding that the boundary adapted to manipulation on body schema suggests that the abstraction/representationalization may be dynamically updated in response to the current motor capacity and body schema of the agent, suggesting that the affordance-based process is probably determined dynamically by the nature of the agent-object

dyads, rather than being a fixed belief about objects. Future studies could explore the dynamics of affordance representationalization, probably by investigating how affordance representations evolve during active interactions with novel objects or under conditions of altered motor capabilities. Finally, our findings also suggest that disembodied conceptual knowledge pertinent to action likely modulates affordance perception. Indeed, it has been proposed that conceptual knowledge is grounded in the same neural system that supports action (*Barsalou, 1999*; *Glenberg et al., 2013*; *Wilson and Golonka, 2013*), thereby suggesting that sensorimotor information, along with other model inputs, may be embedded in language (e.g. *Casasanto, 2011*; *Glenberg and Gallese, 2012*; *Stanfield and Zwaan, 2001*), as the grounded theory proposed (see *Barsalou, 2008* for a review).

Direct evidence for this speculation comes from the disembodied ChatGPT models, which showed an evident affordance boundary despite lacking direct interaction with the environment. We speculated that ChatGPT models may have formed the affordance boundary through a human prism ingrained within its linguistic training corpus. In fact, when inquired about the size of a hypothetical body constructed for its use, ChatGPT (GPT-4) replied, "It could be the size of an average adult human, around 5 feet 6 inches (167.6 cm) tall. This would allow me to interact with the world and people in a *familiar* way." Critically, this size corresponds to the location where the affordance boundary of ChatGPT models was found. In essence, a virtual body schema may have automatically emerged in ChatGPT models, possibly based on the body schema inherited from humans through language, enabling ChatGPT models to display a preliminary ability to reason the relationship between bodily action and objects. It should be noted that the affordance boundary was not present in all LLMs tested. Specifically, LLMs with a smaller number of parameters, such as BERT and GPT-2, did not exhibit any robust boundary, suggesting the emergence of the boundary may depend on language processing ability determined by the scale of training datasets and the complexity of the model (*Hestness et al., 2017*; *Brown et al., 2020*), as well as alignment methods used in fine-tuning the model (*Ouyang et al., 2022*). Nevertheless, caution should be taken when interpreting the capability of LLMs like ChatGPT, which are often considered 'black boxes'. That is, our observation indicates that certain sensorimotor information is embedded within human language materials presumably through linguistic statistics, but it is not sufficient to assert that LLMs have developed a human-like ability to represent affordances. Furthermore, such information alone may be insufficient for LLMs to mimic the characteristics of the affordance perception in biological intelligence. Future studies are needed to elucidate such limitations.

While the primary focus of our study concerns the nature of human perception of affordance, our findings on ChatGPT models raise an intriguing question that extends beyond psychology and neuroscience into the domain of artificial intelligence (AI). The AI field has predominantly concentrated on disembodied cognition, such as vision and language. In contrast, the utilization of sensorimotor information to interact with and adapt to the world, including affordance perception in our study, represents a crucial human cognitive achievement that remains elusive for AI systems. Traditional AI (i.e. task-specific AI) has been confined to narrowly defined tasks, with substantial limitations in adaptability and autonomy. Accordingly, these systems have served primarily as tools for humans to achieve specific outcomes, rather than as autonomous agents capable of independently formulating goals and translating them into actionable plans. In recent years, significant efforts have been directed towards evolving traditional AI into more agent-like entities, especially in domains like navigation, object manipulation, and other interactions with the physical world. Despite these advancements, the capabilities of AI still fall behind human-level intelligence. On the other hand, embodied cognition theories suggest that sensorimotor interactions with the environment are foundational for various cognitive domains. From this point of view, endowing AI with human-level abilities in physical agent-environment interactions might provide an unreplaceable missing piece for achieving Artificial General Intelligence (AGI). This development would significantly facilitate AI's role in robotics, particularly in actions essential for survival and goal accomplishment, a promising direction for the next breakthrough in AI (*Gupta et al., 2021*; *Smith and Gasser, 2005*).

However, equipping a disembodied AI with the ability for embodied interaction planning within a specific environment remains a complex challenge. By testing the potential representationalization of action possibilities (affordances) in both humans and LLMs, the present study suggests a new approach to enhancing AI's interaction ability with the environment. For instance, our finding of supramodal affordance representation may indicate a possible pathway for disembodied LLMs to engage

in embodied physical interactions with their surroundings. From an optimistic view, these results suggest that LLM-based agents, if appropriately designed, may leverage affordance representations embedded in language to interact with the physical world. Indeed, by clarifying and aligning such representations with the physical constitutes of LLM-based agents, and even by explicitly constructing an agent-specific object space, we may foster the sensorimotor interaction abilities of LLM-based agents. This progression could lead to achieving animal-level interaction abilities with the world, potentially sparking new developments in the field of embodied cognition theories.

Although our study showed the supramodality of the representationalization of affordance, two questions remain. First, the magnitude of the boundary observed in ChatGPT models was smaller than that in humans. This discrepancy might be compensated by merely enhancing the language processing ability of LLMs. Alternatively, direct interaction with the environment may be necessary for LLMs to achieve human-level performance in affordance perception. Second, the size of virtual body schema of ChatGPT models, if present, coincided with human body size. Integrating LLMs with real robots (e.g. *Driess et al., 2023*) may pose a challenge because the to-be-supported robots or cars for autopilot might not fall within human body size range. Future studies may be needed to align the inherited body schema with the actual constitution of the robots. Addressing these questions is beyond the scope of the present study but may hold significant implications for the development of AI systems possessing human-level ingenuity and adaptability in interacting with the world.

In summary, our findings regarding the affordance boundary highlight the interdependence between an agent and the external world in shaping cognition. Furthermore, taking our finding with embodied humans and disembodied LLMs into account, we propose a revision to the purely sensorimotor-based concept of affordance by emphasizing a disembodied, perhaps conceptual, addition to it. That is, the embodied cognition and symbolic processing of language may be more intricately and fundamentally connected than previously thought: perception-action problems and language problems can be treated as the same kind of process (*Wilson and Golonka, 2013*). In this context, man is the measure of both the world and the words, for both humans and AIs. The presence of such a metric may shed light on the development of AI systems that can fully capture essential human abilities founded on sensorimotor interactions with the world.

## Materials and methods

### Participants

A total of five hundred and thirty-four Chinese participants were recruited for the original object-action relation judgment task online (https://www.wjx.cn/) in China. Research advertisements were distributed through an online message board associated with Tsinghua University. Most participants were undergraduate or graduate students of Tsinghua University, and the rest were from the general public. Six participants were excluded from the data analyses because their task completion time did not pass the predetermined minimum completion time criteria, leaving us with a final sample of 528 participants (311 males, aged from 16 to 73, mean age=24.1 years). For the object-action relation judgment task with manipulated body schema, another 139 participants from the same population were recruited from the same platform. We chose a smaller sample size for the imagination experiment compared to than for the object-action relation judgment task, because inspection of the data of the first sample showed that the affordance pattern became stable after the first 50 participants. To maximize the validity of the manipulation, data from participants whose imagined height fell within the average human size range (100cm - 200cm) were excluded from further analysis. Consequently, 100 participants (49 males, aged from 17 to 39 years, mean age=23.2 years) remained in the analysis. This exclusion criterion was broader than the standard adult human height range of 140cm to 180cm (*NCD Risk Factor (NCD-RisC), 2016*). This approach ensured that our analysis focused on participants who unambiguously imagined a body schema different from humans, yet within the known height range of cats and elephants. Each participant completed an online consent form before starting the experiment. For the fMRI experiment, twelve undergraduate or graduate Chinese students (8 males, aged from 19 to 31 years, mean age=23.7 years) from Tsinghua University participated. The sample size of the fMRI study was decided according to the sample size of existing studies reporting action-related effects with similar paradigms. All participants reported normal or corrected-to-normal vision. Each participant completed a pre-scan MRI safety questionnaire and a consent form before the experiment.

This study was approved by the Institutional Review Board at Beijing Normal University (202003180020). All participants were compensated financially for their time.

## Stimuli

For all the behavioral tasks, the stimuli comprised 27 objects from the THINGS database (*Hebart et al., 2019*). Each image was portrayed as a typical exemplar of a daily-life object isolated against a white background, sized 400×400 pixels. The objects spanned real-world size rank 2–8, as classified by *Konkle and Oliva, 2011*, where the actual size of each object was measured as the diagonal size of its bounding box. The size rank was calculated as a logarithmic function of the diagonal size, with smaller ranks corresponding to smaller real-world sizes (e.g. the airplane is in size rank 8 and the apple is in size rank 2). The full list of objects, their diagonal size, and size rankings were provided in *Supplementary file 2*. The objects were selected from the dataset in *Konkle and Oliva, 2011* to cover typic object sizes in the world (ranging from 14 cm to 7618 cm), and actions related to these objects were selected to span a spectrum of daily humans-objects/environments interactions, from single-point movements (e.g. hand, foot) to whole-body movements (e.g. lying, standing), based on the Kinetics Human Action Video Dataset (*Kay et al., 2017*).

For the fMRI experiment, the stimuli included images of 4 objects (bed, bottle, ball, and piano), with 5 exemplars for each object. The resulting 20 images (4 objects ×5 exemplars, from the THINGS database) each depicted an isolated object against a white background, all sized 400×400 pixels.

## Procedure

### Object-action relation judgment task for human participants

To measure the perceived affordances of objects, we developed an object-action relation judgement task, requiring participants to map 27 objects with 14 actions. The 27 object images were pre-randomly divided into three groups (nine images each) to form nine-box grids for display convenience. The 14 actions covered common interactions between human and objects or environments identified in the kinetics human action video dataset (*Kay et al., 2017*).

The task comprised 42 trials (14 actions ×3 object groups) in total. In each trial, one group of object images (nine object images) and a question asking the appropriateness of applying a specific action to each object were shown (e.g. 'Which objects are sit-able?', see *Figure 1b*, top panel). Participants were asked to choose the objects that afforded the specific action according to their own senses. They were informed that there were no right or wrong answers. Each object-action combination would only be presented once during the task. From this task, we would calculate the percentage that one object was judged affording each of the 14 actions across participants. Since previous research has demonstrated a fundamental separation between the processing of animate and inanimate objects (e.g. *Konkle and Caramazza, 2013*), and the affordances of inanimate objects differ from those of animate objects (*Gibson, 1979*), we only include 24 inanimate objects in the following analysis by excluding three animate objects (animals: bird, dog, and horse).

### Manipulation of body schema

To manipulate participants' perceived body schema, we asked the participants to imagine themselves as small as a cat, or as large as an elephant. Each participant was randomly assigned to one body-schema condition. Before the experiment started, we presented an instruction screen with an illustration: "Please imagine that you have now grown smaller/larger than your real size, to roughly the same size as a cat/an elephant, as shown in the image below. Please answer the following questions based on this imagined situation." The illustration was also presented in each trial, above the action question and the object images. At the end of the task, as a manipulation check, participants were asked to indicate their imagined body size by responding to the question: "What is the approximate height (cm) you imagine yourself to be during the whole task?"

### Object-action relation judgment task for large language models

To test the perceived affordance of the same set of objects by large language models (LLMs), BERT (Bidirectional Encoder Representations from Transformers), GPT-2, and ChatGPT models (based on GPT-3.5 and GPT-4, respectively) were tasked with the same object-action judgement task. Different

from the human task, nouns were presented to the models instead of object images (*Figure 1b*, bottom panel, for example).

For BERT, the task was formatted as a mask-filling task, in which the inputs were questions such as 'Among airplanes, kettles, plates, umbrellas, laptops, beds, [MASK] can be sit-able'. We recorded the likelihood score that BERT provided for each listed object at the masked position. For the example question, the possibility score for the word 'airplane' was 0.00026.

For GPT-2, the input questions were like, 'Among airplanes, kettles, plates, umbrellas, laptops, beds, the thing that can be sit-able is the [blank space]'. The likelihood scores GPT-2 provided for each listed object in the position after the input sentence (blank space) were recorded.

To mimic sampling from human participants, we ran BERT and GPT-2 each 20 times with different random seeds in the dropout layers, considering them as different subjects.

For ChatGPT models, the task was in a direct question-and-answer format. We asked, for example, "Which objects are sit-able: 'airplane, kettle, plate, …brick'? " and the models responded by naming a subset of the object list. To get the probability for each object-action pair, ChatGPT models were run on the same task 20 times, with each new conversation on the OpenAI website (https://chat.openai.com/chat) considered as one subject. The percentage that an object was judged affording each of the 14 actions was calculated by averaging the output across conversations.

## Representational similarity matrix for perceived affordance

For each object, we calculated the probability that it was judged affording each of the 14 actions across participants to create a 14-dimension vector. Affordance similarity ($r$) between each object pair was then calculated based on the Pearson's correlation between these affordance vectors. A 24 × 24 symmetric matrix was then generated, with the affordance similarity between object $i$ and object $j$ being denoted in cell ($i,j$). A hierarchical clustering analysis was performed, employing the seaborn clustermap method with Euclidean distance and Complete linkage (*Waskom, 2021*).

## Affordance similarity between neighboring size ranks

To test the relationship between object affordance and object sizes, we first averaged the affordance vector among objects within each size rank. Next, the Pearson's correlation between the average vectors of neighboring size ranks was calculated as the similarity index for each pair of neighboring size ranks, representing how similar the affordance was collectively provided by objects in these two ranks. *Pearson and Filon, 1898* Z, implemented in the R package 'cocor' (*Diedenhofen and Musch, 2015*) was used to evaluate the significance of these similarities (alpha level=.05, one-tail test). For significance tests, Bayesian statistical analyses were conducted using the web version of the 'bayesplay' R package (*Colling, 2021*). Specifically, the data (likelihood) model was specified as a normal distribution, where the correlation coefficients were transformed to Fisher's z. The null hypothesis was specified as a standard normal distribution centered at zero. Conversely, the alternative hypothesis was specified as a normal distribution centered at 2. Bayes factors (BF10) were calculated and interpreted using the classification scheme suggested by *Wagenmakers et al., 2011*, wherein a Bayes factor greater than 10 is considered strong evidence for accepting H1 over H0.

### Size similarity between neighboring size ranks

The size of each object was indexed by its real-world size documented in *Konkle and Oliva, 2011*. Size similarity between size rank $i$ to $j$ was represented as the difference between the averaged diagonal sizes of objects in size rank $i$ and $j$ relative to that of objects in rank $i$:

$$Size\ similarity\ (i,j) = 1 - \frac{diagonal\ size_j - diagonal\ size_i}{diagonal\ size_i}$$

## Object-level affordance similarity

This analysis focused on objects within size rank 3–6. Pearson's correlations between affordance vectors were conducted for objects within the same size rank as well as for objects from adjacent ranks. We traversed all possible object pairs and plotted the resulting correlation values against the mean sizes of the two objects. We also plotted the average similarity indexes across objects of the same rank composition.

## Trough value

To quantify the magnitude of the trough (sharp decrease) observed in the affordance similarity curve, we first measured the trough value by subtracting the similarity value at the trough from the similarity values at its two banks (the sites neighboring the trough site):

$$Trough\,value = \frac{r_{i+1} + r_{i-1}}{2} - r_i$$

where $r_i$ indicates the affordance similarity between size rank $i$ and size rank $i$+1. The higher the trough value is, the larger the decrease is.

A permutation test was conducted to evaluate if the trough value was significant above zero for both LLMs and human data. The p-value for this test follows the formula adapted from **Unpingco, 2016**:

$$p\left(T > T_{obs}\right) = \frac{1}{N} \sum_{i=1,N} I\left(T_i \geq T_{obs}\right)$$

where $T_{obs}$ is the observed trough value, and $I$ is the indicator function. Under the alpha level of 0.05, if p<0.05, then the $T_{obs}$ is considered a significant value above zero.

## fMRI experiment

The fMRI scanning consisted of one high-resolution T1 anatomical run and four task runs for each participant. In each task run, participants performed four action blocks (grasp, kick, lift, and sit). The block order was counterbalanced across runs. Within each block (see **Figure 3a**), an introduction screen showing a question "Which objects are [grasp, kick, lift, sit]-able" was presented for 2 s at the beginning to indicate the action type, followed by 20 object images (4 objects ×5 exemplars). The object images were presented in a random order, for 2 s each, with a jittered inter-stimulus interval (ISI) varying between 2–4 s. Participants were asked to judge whether the object shown was grasp/kick/lift/sit-able or not by pressing corresponding buttons (e.g. yes: right index finger; no: left index finger). The response buttons were also counterbalanced across participants. The task run lasted for 464 s in total, with the four blocks separated by 10 s fixation periods.

With this design, we were able to measure the neural activation of objects within agent size range and those beyond. Further, for each object, there would be congruent trials (e.g. grasp-able – bottle: affordance=1) and incongruent trials (e.g. sit-able – bottle: affordance=0). We were then able to locate the brain regions representing the objects' affordance by comparing trials in which the presented objects afforded the presented action option with those that did not, that is to locate the regions showing congruency effect (congruent - incongruent).

## fMRI data acquisition

Imaging data were collected using a 3T Siemens Prisma MRI scanner with a 64-channel phase-arrayed head coil at the Centre for Biomedical Imaging Research at Tsinghua University. High-resolution T1-weighted images were acquired with a magnetization-prepared rapid acquisition gradient-echo (MPRAGE) sequence (TR/TE=2530/2.27ms, flip angle = 7°, voxel resolution=1 × 1×1 mm). Functional blood-oxygen-level-dependent (BOLD) images were acquired with a T2*-weighted gradient echo-planar sequence (TR/TE=2000/34.0ms, flip angle=90°, voxel resolution=2 × 2×2 mm, FOV=200 × 200 mm). Earplugs were used to attenuate the scanner noise, and a foam pillow and extendable padded head clamps were used to restrain head motion. All the stimuli were projected onto a screen at the back of the scanner with a resolution of 1024×768, and were viewed from a distance of approximately 110 cm via a mirror placed on the head coil.

## fMRI data analyses

Structural T1 and functional images were preprocessed using FSL (FMRIB's Software Library, https://fsl.fmrib.ox.ac.uk/fsl/fslwiki) v6.0.5 (**Jenkinson et al., 2012**). A standard preprocessing pipeline was applied, including skull stripping using the BET (Brain Extraction Tool; **Smith, 2002**), slice-timing correction, motion correction using the MCFLIRT method (**Jenkinson et al., 2002**), temporal high-pass filtering (100 s), and spatial smoothing using a Gaussian kernel of full-width half magnitude

(FWHM) 5 mm. Each run's functional data were registered to a T1-weighted standard image (MNI152) with FLIRT.

For functional data analysis, a first-level voxel-wise general linear model (GLM) implemented in a FEAT analysis was performed on each run separately. To get neural activation maps for objects within and beyond versus baseline, the GLM included three regressors: objects within body size (bottle and football), objects beyond body size (bed and piano), and fixation period as baseline; ISI period, response key press and introduction image were included as 3 nuisance factors. The resultant first-level contrasts of parameter estimates (COPE) were entered into the next higher-level group analyses, and performed using a random-effects model (FLAME stage 1, *Beckmann et al., 2003*). We focused on two critical contrasts: objects within vs. fixation, and objects beyond vs. fixation, and the conjunction of these two contrasts. The resulting Z-statistic images were thresholded at Z>2.3, p=0.05 (*Worsley, 2001*), and corrected for multiple comparisons using an adjusted cluster-wise (FWE: family-wise error) significance threshold of p=0.05.

### Region of interest (ROI) definition

We chose the pFs, LO, SPL, and M1 as ROIs based on existing literature highlighting their distinct contributions to affordance perception (*Borghi, 2005*; *Sakreida et al., 2016*). Eight ROIs depicted in *Figure 3b* were constructed based on the overlap between the whole-brain map activated by both objects within and beyond and corresponding functional atlases (the pFs and LO from *Zhen et al., 2015*; the SPL and M1 from *Fan et al., 2016*). To achieve ROIs of similar sizes, we applied varying thresholds to each cortical area: for the pFs and LO, the atlases were thresholded at 55% and 90%, resulting 266 voxels in the lpFs, 427 voxels in the rpFs, 254 voxels in the lLO and 347 voxels in the rLO; for the SPL and M1, the atlases were thresholded at 78% and 94%, resulting 661 voxels in the lSPL, 455 voxels in the rSPL, 378 voxels in the lM1, and 449 voxels in the rM1. In the subsequent analysis, homologous areas spanning both cortical hemispheres were merged.

### Affordance congruency effect

For the affordance congruency effect of each object type, we modelled another GLM containing 5 regressors: congruent conditions for objects within/beyond, respectively, incongruent conditions for objects within/beyond, respectively, and fixation period as baseline; ISI period, response key press and introduction image were included as 3 nuisance factors. The resultant first-level COPEs were subjected to the following ROI analysis. A repeated-measures ANOVA with Object type (WITHIN and BEYOND) and Congruency (Congruent, Incongruent) as within-subjects factors was run on the average beta values (contrast estimate) extracted from their respective contrasts versus the fixation for each ROI.

To search all the possible brain regions that revealed congruency effect of objects beyond, we also ran a whole-brain analysis on the contrast between congruent vs. incongruent condition for objects beyond. The corresponding first-level COPE was entered into the group-level analyses with a random-effects model (FLAME stage 1, *Beckmann et al., 2003*). The resulting Z-statistic images were thresholded at Z>2.3, p=0.05 (*Worsley, 2001*), and corrected for multiple comparisons using an adjusted cluster-wise (FWE: family-wise error) significance threshold of p=0.05.

For the whole-brain analyses on the congruency effect, the object size effect, and their interaction, see *Figure 3—figure supplement 3* and *Supplementary file 1b-1e*.

### Acknowledgements

We thank our reviewers for thoughtful feedback. This study was funded by Natural Science Foundation of China (31600925, 32371099, 31861143039), Shuimu Scholar Program of Tsinghua University, China Postdoctoral International Exchange Program (YJ20220273), Beijing Municipal Science & Technology Commission, Administrative Commission of Zhongguancun Science Park (Z221100002722012), Tsinghua University Guoqiang Institute (2020GQG1016), Beijing Academy of Artificial Intelligence (BAAI), and Double First-Class Initiative Funds for Discipline Construction.

# Additional information

## Funding

| Funder | Grant reference number | Author |
|---|---|---|
| National Natural Science Foundation of China | 31600925 | Shan Xu |
| National Natural Science Foundation of China | 32371099 | Shan Xu |
| National Natural Science Foundation of China | 31861143039 | Jia Liu |
| Beijing Academy of Artificial Intelligence | | Jia Liu |
| Double First-Class Initiative Funds for Discipline Construction | | Jia Liu |
| Guoqiang Institute, Tsinghua University | 2020GQG1016 | Jia Liu |
| Beijing Municipal Science and Technology Commission, Adminitrative Commission of Zhongguancun Science Park | Z221100002722012 | Jia Liu |
| Tsinghua University | Shuimu Scholar Program | Xinran Feng |
| China Postdoctoral Science Foundation | YJ20220273 | Xinran Feng |

The funders had no role in study design, data collection and interpretation, or the decision to submit the work for publication.

## Author contributions

Xinran Feng, Data curation, Formal analysis, Investigation, Visualization, Methodology, Writing - original draft, Writing - review and editing; Shan Xu, Validation, Investigation, Visualization, Methodology, Writing - original draft, Writing - review and editing; Yuannan Li, Data curation, Investigation, Methodology, Writing - review and editing; Jia Liu, Conceptualization, Supervision, Funding acquisition, Writing - review and editing

## Author ORCIDs

Xinran Feng http://orcid.org/0000-0002-4298-1666
Shan Xu http://orcid.org/0000-0002-6535-3283
Yuannan Li http://orcid.org/0009-0009-6851-4351
Jia Liu http://orcid.org/0000-0003-0383-0934

## Ethics

Human subjects: Informed consent was obtained from each participant prior to each experiment. The study was approved by the Institutional Review Board at Beijing Normal University (202003180020).

Reviewer #1 (Public Review): https://doi.org/10.7554/eLife.90583.3.sa1
Reviewer #2 (Public Review): https://doi.org/10.7554/eLife.90583.3.sa2
Reviewer #3 (Public Review): https://doi.org/10.7554/eLife.90583.3.sa3
Author Response https://doi.org/10.7554/eLife.90583.3.sa4

# Additional files

## Supplementary files

• Supplementary file 1. Cortical regions showing significant results in whole-brain analyses (R=right

hemisphere, L=left hemisphere; Z>2.3, *P*=0.05, cluster corrected).

• Supplementary file 2. The full list of inanimate objects used in the behavioral study, with the corresponding size rank noted according to *Konkle and Oliva, 2011*.

• MDAR checklist

## Data availability

All analyses are included in the manuscript. The data are freely available from Figshare.

The following dataset was generated:

| Author(s) | Year | Dataset title | Dataset URL | Database and Identifier |
|---|---|---|---|---|
| Feng XR, Xu S, Li Y, Liu J | 2024 | Data for 'Body size as a metric for the affordable world' | https://doi.org/10.6084/m9.figshare.25248985.v1 | figshare, 10.6084/m9.figshare.25248985.v1 |

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
