## [Editor Report · eLife assessment]

This paper presents **valuable** findings that shed light on the mental organisation of knowledge about real-world objects. It provides diverse, if **incomplete** and tentative, evidence from behaviour, brain, and large language models that this knowledge is divided categorically between relatively small objects (closer to the relevant scale for direct manipulation) and larger objects (further from the typical scope of human affordances for action).

---

## [Referee Report · Reviewer #1 (Public Review)]

Ps observed 24 objects and were asked which afforded particular actions (14 action types). Affordances for each object were represented by a 14-item vector, values reflecting the percentage of Ps who agreed on a particular action being afforded by the object. An affordance similarity matrix was generated which reflected similarity in affordances between pairs of objects. Two clusters emerged, reflecting correlations between affordance ratings in objects smaller than body size and larger than body size. These clusters did not correlate themselves. There was a trough in similarity ratings between objects ~105 cm and ~130 cm, arguably reflecting the body size boundary. The authors subsequently provide some evidence that this clear demarcation is not simply an incidental reflection of body size, but likely causally related. This evidence comes in the flavour of requiring Ps to imagine themselves as small as a cat or as large as an elephant and showing a predicted shift in the affordance boundary. The manuscript further demonstrates that ChatGPT (theoretically interesting because it's trained on language alone without sensorimotor information; trained now on words rather than images) showed a similar boundary.

The authors also conducted a small MRI study task where Ps decide whether a probe action was affordable (graspable?) and created a congruency factor according to the answer (yes/no). There was an effect of congruency in posterior fusiform and superior parietal lobule for objects within body size range, but not outside. No effects in LOC or M1.

The major strength of this manuscript in my opinion is the methodological novelty. I felt the correlation matrices were a clever method for demonstrating these demarcations, the imagination manipulation was also exciting, and the ChatGPT analysis provided excellent food for thought. These findings are important for our understanding of the interactions between action and perception, and hence for researchers from a range of domains of cognitive neuroscience.

The major element that limits conclusions is that an MRI study with 12 P in this context can really only provide pilot data. Certainly the effects are not strong enough for 12 P to generate much confidence. The others of my concerns have been addressed in the revision.

---

## [Referee Report · Reviewer #2 (Public Review)]

Summary

In this work, the authors seek to test a version of an old idea, which is that our perception of the world and our understanding of the objects in it are deeply influenced by the nature of our bodies and the kinds of behaviours and actions that those objects afford. The studies presented here muster three kinds of evidence for a discontinuity in the encoding of objects, with a mental "border" between objects roughly of human body scale or smaller, which tend to relate to similar kinds of actions that are yet distinct from the kinds of actions implied by human-or-larger scale objects. This is demonstrated through observers' judgments of the kinds of actions different objects afford; through similar questioning of AI large-language models (LLMs); and through a neuroimaging study examining how brain regions implicated in object understanding make distinctions between kinds of objects at human and larger-than-human scales.

Strengths

The authors address questions of longstanding interest in the cognitive neurosciences -- namely how we encode and interact with the many diverse kinds of objects we see and use in daily life. A key strength of the work lies in the application of multiple approaches. Examining the correlations among kinds of objects, with respect to their suitability for different action kinds, is novel, as are the complementary tests of judgments made by LLMs. The authors include a clever manipulation in which participants are asked to judge action-object pairs, having first adopted the imagined size of either a cat or an elephant, showing that the discontinuity in similarity judgments effectively moved to a new boundary closer to the imagined scale than the veridical human scale. The dynamic nature of the discontinuity hints that action affordances may be computed dynamically, "on the fly", during actual action behaviours with objects in the real world.

Weaknesses

A limitation of the tests of LLMs may be that it is not always known what kinds of training material was used to build these models, leading to a possible "black box" problem. Further, presuming that those models are largely trained on previous human-written material, it may not necessarily be theoretically telling that the "judgments" of these models about action-object pairs shows human-like discontinuities. Indeed, verbal descriptions of actions are very likely to mainly refer to typical human behaviour, and so the finding that these models demonstrate an affordance discontinuity may simply reflect those statistics, rather than providing independent evidence for affordance boundaries.

The relatively small sample size of the brain imaging experiment, and some design features (such as the task participants performed, and the relatively narrow range of objects tested) provide some limits on the extent to which it can be taken as support for the authors' claims.

---

## [Referee Report · Reviewer #3 (Public Review)]

Summary:

Feng et al. test the hypothesis that human body size constrains the perception of object affordances, whereby only objects that are smaller than the body size will be perceived as useful and manipulable parts of the environment, whereas larger objects will be perceived as "less interesting components."

To test this idea, the study employs a multi-method approach consisting of three parts:

In the first part, human observers classify a set of 24 objects that vary systematically in size (e.g., ball, piano, airplane) based on 14 different affordances (e.g., sit, throw, grasp). Based on the average agreement of ratings across participants, the authors compute the similarity of affordance profiles between all object pairs. They report evidence for two homogenous object clusters that are separated based on their size with the boundary between clusters roughly coinciding with the average human body size. In follow-up experiments, the authors show that this boundary is larger/smaller in separate groups of participants who are instructed to imagine themselves as an elephant/cat.

In the second part, the authors ask different large language models (LLMs) to provide ratings for the same set of objects and affordances and conduct equivalent analyses on the obtained data. Some, but not all, of the models produce patterns of ratings that appear to show similar boundary effects, though less pronounced and at a different boundary size than in humans.

In the third part, the authors conduct an fMRI experiment. Human observers are presented with four different objects of different sizes and asked if these objects afford a small set of specific actions. Affordances are either congruent or incongruent with objects. Contrasting brain activity on incongruent trials against brain activity on congruent trials yields significant effects in regions within the ventral and dorsal visual stream, but only for small objects and not for large objects.

The authors interpret their findings as support for their hypothesis that human body size constrains object perception. They further conclude that this effect is cognitively penetrable, and only partly relies on sensorimotor interaction with the environment (and partly on linguistic abilities).

Strengths:

The authors examine an interesting and relevant question and articulate a plausible (though somewhat underspecified) hypothesis that certainly seems worth testing. Providing more detailed insights into how object affordances shape perception would be highly desirable. Their method of analyzing similarity ratings between sets of objects seems useful and the multi-method approach is original and interesting.

Weaknesses:

The study presents several shortcomings that clearly weaken the link between the obtained evidence and the drawn conclusions. Below I outline my concerns in no particular order:

(1) It is not entirely clear to me what the authors are proposing and to what extent the conducted work actually speaks to this. For example, in the introduction, the authors write that they seek to test if body size serves not merely as a reference for object manipulation but also "plays a pivotal role in shaping the representation of objects." This motivation seems rather vague motivation and it is not clear to me how it could be falsified.

Overall, the lack of theoretical precision makes it difficult to judge the appropriateness of the approaches and the persuasiveness of the obtained results. I would strongly suggest clarifying the theoretical rationale and explaining in more detail how the chosen experiments allow them to test falsifiable predictions.

(2) The authors used only a very small set of objects and affordances in their study and they do not describe in sufficient detail how these stimuli were selected. This renders the results rather exploratory and clearly limits their potential to discover general principles of human perception. Much larger sets of objects and affordances and explicit data-driven approaches for their selection would provide a more convincing approach and allow the authors to rule out that their results are just a consequence of the selected set of objects and actions.

(3) Relatedly, the authors could be more thorough in ruling out potential alternative explanations. Object size likely correlates with other variables that could shape human similarity judgments and the estimated boundary is quite broad (depending on the method, either between 80 and 150 cm or between 105 to 130 cm). More precise estimates of the boundary and more rigorous tests of alternative explanations would add a lot to strengthen the authors' interpretation.

(4) While I appreciate the manipulation of imagined body size, as a clever way to solidify the link between body size and affordance perception, I find it unfortunate that it is implemented in a between-subjects design, as this clearly leaves open the possibility of pre-existing differences between groups. I certainly disagree with the authors' statement that their findings suggest "a causal link between body size and affordance perception."

(5) The use of LLMs in the current study is not clearly motivated and I find it hard to understand what exactly the authors are trying to test through their inclusion. As it currently stands, I find it hard to discern how the presence of perceptual boundaries in LLMs could constitute evidence for affordance-based perception.

(6) Along the same lines, the fMRI study also provides little evidence to support the authors' claims. The use of congruency effects as a way of probing affordance perception is not well motivated. Importantly (and related to comment 2 above), the very small set of objects and affordances in this experiment heavily complicates any conclusions about object size being the crucial variable determining the occurrence of congruency effects.

Overall, I consider the main conclusions of the paper to be far beyond the reported data. Articulating a clearer theoretical framework with more specific hypotheses as well as conducting more principled analyses on more comprehensive data sets could help the authors obtain stronger tests of their ideas.

---

## [Author Response]

The following is the authors’ response to the current reviews.

**Responses to the reviewers**

We thank the editor and reviewers for their insightful feedback and valuable suggestions on our revised manuscript. In this reply, we provided further clarifications and made changes accordingly. Reviewers’ comments are in bold, and our responses are immediately below. Changes in the main text are presented in italics, accompanied by the specific line numbers in the revised manuscript where these changes can be found. Below, we respond to each reviewer’s comments in turn.

**Reviewer #1 (Public Review):**
Ps observed 24 objects and were asked which afforded particular actions (14 action types). Affordances for each object were represented by a 14-item vector, values reflecting the percentage of Ps who agreed on a particular action being afforded by the object. An affordance similarity matrix was generated which reflected similarity in affordances between pairs of objects. Two clusters emerged, reflecting correlations between affordance ratings in objects smaller than body size and larger than body size. These clusters did not correlate themselves. There was a trough in similarity ratings between objects ~105 cm and ~130 cm, arguably reflecting the body size boundary. The authors subsequently provide some evidence that this clear demarcation is not simply an incidental reflection of body size, but likely causally related. This evidence comes in the flavour of requiring Ps to imagine themselves as small as a cat or as large as an elephant and showing a predicted shift in the affordance boundary. The manuscript further demonstrates that ChatGPT (theoretically interesting because it's trained on language alone without sensorimotor information; trained now on words rather than images) showed a similar boundary.The authors also conducted a small MRI study task where Ps decide whether a probe action was affordable (graspable?) and created a congruency factor according to the answer (yes/no). There was an effect of congruency in posterior fusiform and superior parietal lobule for objects within body size range, but not outside. No effects in LOC or M1.The major strength of this manuscript in my opinion is the methodological novelty. I felt the correlation matrices were a clever method for demonstrating these demarcations, the imagination manipulation was also exciting, and the ChatGPT analysis provided excellent food for thought. These findings are important for our understanding of the interactions between action and perception, and hence for researchers from a range of domains of cognitive neuroscience.The major element that limits conclusions is that an MRI study with 12 P in this context can really only provide pilot data. Certainly the effects are not strong enough for 12 P to generate much confidence. The others of my concerns have been addressed in the revision.
**Reviewer #1 (Recommendations For The Authors):**
I think that the authors need to mention in the abstract that the MRI study constitutes a small pilot.

Response: We appreciate the reviewer’s positive evaluation and constructive suggestions. In response to the concern about the limited number of participants in the fMRI study, we fully acknowledge the implications this has on the generalizability and robustness of our findings related to the congruency effect. To clarity, we have explicitly stated its preliminary nature of the MRI study in the abstract [line 22]: “A subsequent fMRI experiment offered preliminary evidence of affordance processing exclusively for objects within the body size range, but not for those beyond.”

**Reviewer #2 (Public Review):**
SummaryIn this work, the authors seek to test a version of an old idea, which is that our perception of the world and our understanding of the objects in it are deeply influenced by the nature of our bodies and the kinds of behaviours and actions that those objects afford. The studies presented here muster three kinds of evidence for a discontinuity in the encoding of objects, with a mental "border" between objects roughly of human body scale or smaller, which tend to relate to similar kinds of actions that are yet distinct from the kinds of actions implied by human-or-larger scale objects. This is demonstrated through observers' judgments of the kinds of actions different objects afford; through similar questioning of AI large-language models (LLMs); and through a neuroimaging study examining how brain regions implicated in object understanding make distinctions between kinds of objects at human and larger-than-human scales.StrengthsThe authors address questions of longstanding interest in the cognitive neurosciences -- namely how we encode and interact with the many diverse kinds of objects we see and use in daily life. A key strength of the work lies in the application of multiple approaches. Examining the correlations among kinds of objects, with respect to their suitability for different action kinds, is novel, as are the complementary tests of judgments made by LLMs. The authors include a clever manipulation in which participants are asked to judge action-object pairs, having first adopted the imagined size of either a cat or an elephant, showing that the discontinuity in similarity judgments effectively moved to a new boundary closer to the imagined scale than the veridical human scale. The dynamic nature of the discontinuity hints that action affordances may be computed dynamically, "on the fly", during actual action behaviours with objects in the real world.WeaknessesA limitation of the tests of LLMs may be that it is not always known what kinds of training material was used to build these models, leading to a possible "black box" problem. Further, presuming that those models are largely trained on previous human-written material, it may not necessarily be theoretically telling that the "judgments" of these models about action-object pairs shows human-like discontinuities. Indeed, verbal descriptions of actions are very likely to mainly refer to typical human behaviour, and so the finding that these models demonstrate an affordance discontinuity may simply reflect those statistics, rather than providing independent evidence for affordance boundaries.The relatively small sample size of the brain imaging experiment, and some design features (such as the task participants performed, and the relatively narrow range of objects tested) provide some limits on the extent to which it can be taken as support for the authors' claims.

Response: We thank the reviewer for the positive evaluation and the constructive comments. We agree that how LLMs work is a “black box”, and thus it is speculative to assume them to possess any human-like ability, because, as the reviewer pointed out, “these models demonstrate an affordance discontinuity may simply reflect those statistics.” Indeed, our manuscript has expressed a similar idea [line 338]: “We speculated that ChatGPT models may have formed the affordance boundary through a human prism ingrained within its linguistic training corpus.” That is, our intention was not to suggest that such information could replace sensorimotor-based interaction or achieve human-level capability, but rather to highlight that embodied interaction is necessary. Additionally, the scope of the present study does not extend to elucidating the mechanisms behind LLMs’ resemblance of affordance boundary, whether through statistical learning or actual comprehension. To clarify this point, in the revised manuscript, we have clarified that the mechanisms underlying the observed affordance boundary in LLMs may be different from human cognitive processes, and advocated future studies to explore this possibility [line 415]: “Nevertheless, caution should be taken when interpreting the capability of LLMs like ChatGPT, which are often considered “black boxes.” That is, our observation indicates that certain sensorimotor information is embedded within human language materials presumably through linguistic statistics, but it is not sufficient to assert that LLMs have developed a human-like ability to represent affordances. Furthermore, such information alone may be insufficient for LLMs to mimic the characteristics of the affordance perception in biological intelligence. Future studies are needed to elucidate such limitation.”

Regarding the concern about the models’ results not “providing independent evidence for affordance boundaries”, our objective in employing LLMs was to explore if an affordance boundary could emerge from conceptual knowledge without direct sensorimotor experience, rather than to validate the existence of the affordance boundary per se.

As for the concern about the limitations imposed by the small sample size and certain design features of our brain imaging experiment, please see our reply to Reviewer #1.

**Reviewer #3 (Public Review):**
Summary:Feng et al. test the hypothesis that human body size constrains the perception of object affordances, whereby only objects that are smaller than the body size will be perceived as useful and manipulable parts of the environment, whereas larger objects will be perceived as "less interesting components."To test this idea, the study employs a multi-method approach consisting of three parts:In the first part, human observers classify a set of 24 objects that vary systematically in size (e.g., ball, piano, airplane) based on 14 different affordances (e.g., sit, throw, grasp). Based on the average agreement of ratings across participants, the authors compute the similarity of affordance profiles between all object pairs. They report evidence for two homogenous object clusters that are separated based on their size with the boundary between clusters roughly coinciding with the average human body size. In follow-up experiments, the authors show that this boundary is larger/smaller in separate groups of participants who are instructed to imagine themselves as an elephant/cat.In the second part, the authors ask different large language models (LLMs) to provide ratings for the same set of objects and affordances and conduct equivalent analyses on the obtained data. Some, but not all, of the models produce patterns of ratings that appear to show similar boundary effects, though less pronounced and at a different boundary size than in humans.In the third part, the authors conduct an fMRI experiment. Human observers are presented with four different objects of different sizes and asked if these objects afford a small set of specific actions. Affordances are either congruent or incongruent with objects. Contrasting brain activity on incongruent trials against brain activity on congruent trials yields significant effects in regions within the ventral and dorsal visual stream, but only for small objects and not for large objects.The authors interpret their findings as support for their hypothesis that human body size constrains object perception. They further conclude that this effect is cognitively penetrable, and only partly relies on sensorimotor interaction with the environment (and partly on linguistic abilities).Strengths:The authors examine an interesting and relevant question and articulate a plausible (though somewhat underspecified) hypothesis that certainly seems worth testing. Providing more detailed insights into how object affordances shape perception would be highly desirable. Their method of analyzing similarity ratings between sets of objects seems useful and the multi-method approach is original and interesting.Weaknesses:The study presents several shortcomings that clearly weaken the link between the obtained evidence and the drawn conclusions. Below I outline my concerns in no particular order:(1) It is not entirely clear to me what the authors are proposing and to what extent the conducted work actually speaks to this. For example, in the introduction, the authors write that they seek to test if body size serves not merely as a reference for object manipulation but also "plays a pivotal role in shaping the representation of objects." This motivation seems rather vague motivation and it is not clear to me how it could be falsified.Overall, the lack of theoretical precision makes it difficult to judge the appropriateness of the approaches and the persuasiveness of the obtained results. I would strongly suggest clarifying the theoretical rationale and explaining in more detail how the chosen experiments allow them to test falsifiable predictions.(2) The authors used only a very small set of objects and affordances in their study and they do not describe in sufficient detail how these stimuli were selected. This renders the results rather exploratory and clearly limits their potential to discover general principles of human perception. Much larger sets of objects and affordances and explicit data-driven approaches for their selection would provide a more convincing approach and allow the authors to rule out that their results are just a consequence of the selected set of objects and actions.(3) Relatedly, the authors could be more thorough in ruling out potential alternative explanations. Object size likely correlates with other variables that could shape human similarity judgments and the estimated boundary is quite broad (depending on the method, either between 80 and 150 cm or between 105 to 130 cm). More precise estimates of the boundary and more rigorous tests of alternative explanations would add a lot to strengthen the authors' interpretation.(4) While I appreciate the manipulation of imagined body size, as a clever way to solidify the link between body size and affordance perception, I find it unfortunate that it is implemented in a between-subjects design, as this clearly leaves open the possibility of pre-existing differences between groups. I certainly disagree with the authors' statement that their findings suggest "a causal link between body size and affordance perception."(5) The use of LLMs in the current study is not clearly motivated and I find it hard to understand what exactly the authors are trying to test through their inclusion. As it currently stands, I find it hard to discern how the presence of perceptual boundaries in LLMs could constitute evidence for affordance-based perception.(6) Along the same lines, the fMRI study also provides little evidence to support the authors' claims. The use of congruency effects as a way of probing affordance perception is not well motivated. Importantly (and related to comment 2 above), the very small set of objects and affordances in this experiment heavily complicates any conclusions about object size being the crucial variable determining the occurrence of congruency effects.Overall, I consider the main conclusions of the paper to be far beyond the reported data. Articulating a clearer theoretical framework with more specific hypotheses as well as conducting more principled analyses on more comprehensive data sets could help the authors obtain stronger tests of their ideas.

Response: We appreciate the insightful inquiries regarding our manuscript. Below, we explained the theoretical motivation and rationale of each part of our experiments.

In response to the reviewer’s insights, we have modified the expression “plays a pivotal role in shaping the representation of objects” in the revised manuscript and have restated the general question of our study in the introduction. Our motivation is on the long-lasting debate over the representation versus direct perception of affordance, specifically examining the “representationalization” of affordance. That is, we tested whether object affordance simply covaried directly with continuous constraints such as object size, a perspective aligned with the representation-free (direct perception) view, or whether affordance became representationalized, adhering to the representation-based view, constrained by body size. Such representationalization would generate a categorization between objects that are affordable and the environment that exceeds affordance.

To test these hypotheses, we first delineated the affordance of various objects. We agree with the reviewer that in this step a broader selection of objects and actions could mitigate the risk of our results being influenced by the specific selection of objects and actions. However, our results are unlikely to be biased, because our selection was guided by two key criteria, rather than being arbitrary. First, the objects were selected from the dataset in Konkle and Oliva's study (2011), which systematically investigated object size’ impact on object recognition, thus providing a well-calibrated range of sizes (i.e., from 14 cm to 7,618 cm) reflective of real-world objects. Second, the selected actions covered a wide range of daily humans-objects/environments interactions, from single-point movements (e.g., hand, foot) to whole-body movements (e.g., lying, standing) based on the kinetics human action video dataset (Kay et al., 2017). Thus, this set of objects and actions is a representative sampling of typical human experiences.

Upon demonstrating a trough in perceived affordance similarity, we recognized the location of the affordance boundary coincidentally fell within the range of human body size. We agree with the reviewer that this observation of the coincidence between body size and the location of boundary alone is not sufficient for a mechanistic explanation, because variables co-varying with object sizes might also generate this coincidence. The identification of a more precise location for the boundary unlikely rules out alternative explanations of this kind. To establish a causal link between body size and the affordance boundary, we opted for a direct manipulation of body sizes through imagination, while keeping all other variables constant across conditions. This approach allowed us to examine whether and how the affordance boundary shifts in response to body size changes.

Regarding the between-subjects design of the imagination experiment, we wish to clarify that this design aimed to prevent carryover effects. Although a within-subjects design indeed is more sensitive in detecting manipulation effects by accounting for subject variability, it risks contamination across conditions. Specifically, transitioning immediately between different imagined body sizes poses a challenge, and sequential participation could induce undesirable response strategies, such as deliberately altering responses to the same objects in different conditions. The between-subjects design, which susceptible to participant variability (e.g., “pre-existing differences between groups” suggested by the reviewer), avoids such contamination. In addition, we employed random assignment of participants to different conditions (cat-size versus elephant-size).

The body imagination experiment provided causal evidence of an embodied discontinuity, suggesting the boundary is tied to the agent’s motor capacity, rather than amodal sources. The LLMs experiment then sought to test a prediction from the embodied theories of cognition: the supramodality of object perception. Especially, we asked whether the embodied discontinuity is supramodally accessible, using LLMs to assess whether affordance perception discretization is supramodally accessible beyond the sensorimotor domain through linguistic understanding. From this perspective, our LLM experiment was employed not to affirm affordance-based perception but to examine and support a prediction by the embodied theories of cognition.

Finally, our preliminary fMRI study aimed to conceptually replicate the perceptual discontinuity and explore it neural correlates using a subset of objects and actions from the behaviour experiments. This approach was chosen to achieve stable neural responses and enhance study power, employing the congruent effect (congruent - incongruent) as a metric for affordance processing (e.g., Kourtis et al., 2018), which reflects facilitated responses when congruent with objects’ affordances (e.g., Ellis & Tucker, 2000). Nevertheless, we recognize the limitation of a relatively small sample sizes, for details please see our reply to the reviewer #1.

In summary, our findings contribute to the discourse on computationalism’s representation concept and influence of these representations, post-discretization, on processes beyond the sensorimotor domain. We hope that these additional explanations and revisions effectively address the concerns raised and demonstrate our commitment to enhancing the quality of our work in light of your valuable feedback. By acknowledging these limitations and directions for future research, we hope to further the discourse on affordance perception and embodied cognition.

References

Ellis, R., & Tucker, M. (2000). Micro‐affordance: The potentiation of components of action by seen objects. British Journal of Psychology, 91(4), 451-471.

Kay, W., Carreira, J., Simonyan, K., Zhang, B., Hillier, C., Vijayanarasimhan, S., ... & Zisserman, A. (2017). The kinetics human action video dataset. arXiv preprint arXiv:1705.06950.

Konkle, T., & Oliva, A. (2011). Canonical visual size for real-world objects. Journal of Experimental Psychology: human perception and performance, 37(1), 23.

Kourtis, D., Vandemaele, P., & Vingerhoets, G. (2018). Concurrent cortical representations of function-and size-related object affordances: an fMRI study. Cognitive, Affective, & Behavioral Neuroscience, 18, 1221-1232.

The following is the authors’ response to the original reviews.

**Responses to the reviewers**

We deeply appreciate the reviewers’ comments. In response to the concerns raised, we have revised the manuscript accordingly. Below we address each of the reviewers’ comments in turn. Reviewers’ comments are in bold, and our responses are immediately below. Changes in the main text are presented in italics, followed by corresponding page and line numbers in the revised manuscript. We also highlighted tracks of change in the revised manuscript.

**Reviewer #1 (Public Review):**
(1) The main behavioural work appears well-powered (>500 Ps). This sample reduces to 100 for the imagination study, after removing Ps whose imagined heights fell within the human range (100-200 cm). Why 100-200 cm? 100 cm is pretty short for an adult. Removing 80% of data feels like conclusions from the imagination study should be made with caution.

R1: Sorry for the confusion. We did not remove 80% of the participants; instead, a separate sample of participants was recruited in the imagination experiment. The size of this sample (100 participants) was indeed smaller than the first experiment (528 participants), because the first experiment was set for exploratory purposes and was designed to be over-powered. Besides, inspection of the data of the first sample showed that the affordance pattern became stable after the first 50 participants. We explained this consideration in the revised manuscript:

(p 21, ln 490) “…, another one hundred and thirty-nine participants from the same population were recruited from the same platform. We chose a smaller sample size for the imagination experiment compared to that for the object-action relation judgement task, because inspection of the data of the first sample showed that the affordance pattern became stable after the first 50 participants.”

The average adult human height ranges from 140-170 cm for women and 150180 cm for men (NCD-RisC, 2016). Accordingly, the criterion of 100-200 cm covered this range and was set to ensure that participants unambiguously imagined a body schema different from that of human, as the tallest domestic cat below 100 cm according to the Guinness World Records and an elephant above 200 cm according to Crawley et al. (2017). We clarified these considerations in the revised manuscript:

(p 21, ln 494) “To maximize the validity of the manipulation, data from participants whose imagined height fell within the average human size range (100cm - 200cm) were excluded from further analysis. Consequently, 100 participants (49 males, aged from 17 to 39 years, mean age = 23.2 years) remained in the analysis. This exclusion criterion was broader than the standard adult human height range of 140cm to 180cm (NCD-RisC, 2016). This approach ensured that our analysis focused on participants who unambiguously imagined a body schema different from humans, yet within the known height range of cats and elephants.”

In addition, we also reanalysed the data with a more conservative criterion of 140cm to 180cm, and the results remained.

(2) There are only 12 Ps in the MRI study, which I think should mean the null effects are not interpreted. I would not interpret these data as demonstrating a difference between SPL and LOC/M1, but rather that some analyses happened to fall over the significance threshold and others did not.

R2: We would like to clarify that the null hypothesis of this fMRI study is the lack of two-way interaction between object size and object-action congruency, which was rejected by the observed significant interaction. That is, the interpretation of the present study did not rely on accepting any null effect.

Having said this, we admit that the fMRI experiment is exploratory and the sample size is small (12 participants), which might lead to low power in estimating the affordance effect. In the revision, we acknowledge this issue explicitly:

(p 16, ln 354) “…, supporting the idea that affordance is typically represented only for objects within the body size range. While it is acknowledged that the sample size of the fMRI study was small (12 participants), necessitating cautious interpretation of its results, the observed neural-level affordance discontinuity is notable. That is, qualitative differences in neural activity between objects within the affordance boundary and those beyond replicated our behavioral findings. This convergent evidence reinforced our claim that objects were discretized into two broad categories along the continuous size axis, with affordance only being manifested for objects within the boundary.”

(3) I found the MRI ROI selection and definition a little arbitrary and not really justified, which rendered me even more cautious of the results. Why these particular sensory and motor regions? Why M1 and not PMC or SMA? Why SPL and not other parietal regions? Relatedly, ROIs were defined by thresholding pF and LOC at "around 70%" and SPL and M1 "around 80%", and it is unclear how and why these (different) thresholds were determined.

R3: Our selection of these specific sensory and motor regions was based on prior literature reporting their distinct contribution to affordance perception (e.g., Borghi, 2005; Sakreida et al., 2016). The pFs was chosen as a representative region of the ventral visual stream, involved in object identification and classification, and the SPL was chosen as a representative region of the dorsal visual stream, involved in object perception and manipulation. The primary motor cortex (M1) has also been reported involved in affordance processing (e.g., McDannald et al., 2018), and we chose this region to probe the affordance congruency effect in the motor execution stage of the sense-think-act pathway. We did not choose the premotor cortex (PMC) and the supplementary motor area (SMA) because they were proposedly also involved in processes beyond motor execution (e.g., Hertrich et al., 2016; Kantak et al., 2012), and if any effect was observed, one cannot exclusively attribute the effect to motor execution. As for the parietal regions, our choice of the SPL not IPL/IPS is based on the meta-analysis of affordance processing areas where only the SPL shows consistent activation for both stable and variable affordances (Sakreida et al., 2016). We chose the SPL to capture effects on either type of affordances. In revision, we explained these considerations in the revised manuscript:

(p 14, ln 280) “In addition to the pFs and SPL, we also examined the congruency effect in the lateral occipital cortex (LO), which is involved in object representation (e.g., Grill-Spector et al., 2000; Konkle & Caramazza, 2013) and provides inputs to both the pFs and SPL (Hebart et al., 2018). Meanwhile, the primary motor cortex (M1), which receives inputs from the dorsal stream (Vainio & Ellis, 2020), is involved in affordance processing (e.g., McDannald et al., 2018) and action executions(Binkofski et al., 2002).”

(p 29, ln 684) “We chose the pFs, LO, SPL, and M1 as ROIs based on existing literature highlighting their distinct contributions to affordance perception (Borghi,2005; Sakreida et al., 2016).”

Regarding ROI thresholding, we apologize for the lack of clarity in reporting the thresholds in the original manuscript. The thresholds were different between ventral regions (from Zhen et al., 2015) and dorsal regions (from Fan et al., 2016) because they are from two different atlases. The former was constructed by probability maps of task-state fMRI activity during localizer contrast with stationary images and the latter by a parcellation of the brain's functional connectivity; therefore, the numerical values in these two atlases are not comparable. To extract ROIs with comparable sizes, we selected a threshold of 55% for the pFs, 90% for the LO, 78% for the SPL, and 94% for the M1 in the original manuscript.

To rule out the possibility that the results were distorted by the specific choice of thresholds, we re-ran the analysis with a threshold 80% for all ROIs (resulting in 456 voxels in the lpFs, 427 voxels in the rpFs, 1667 voxels in the lLO, 999 voxels in the rLO, 661 voxels in the lSPL, 310 voxels in the rSPL, 231 voxels in the lM1, and 327 voxels in the rM1) with the 2-by-2 repeated-measures ANOVA. Our results remained the same qualitatively. A significant interaction between object type and congruency was observed in the pFs (F(1,11) = 24.87, p <.001, η�2=.69) and SPL (F(1,11) = 14.62, p = .003, η�2=.57). The simple effect analysis revealed the congruency effect solely for objects within body size range (pFs: p = .003; SPL: p <.001), not for objects beyond (ps >.30). For the M1 and LO, neither significant main effects (ps >.11) nor interactions were found (ps >.20).

We clarified our choice of thresholds in the methods section in the revised manuscript:

(p 29, ln 686) “Eight ROIs depicted in Fig. 3b were constructed based on the overlap between the whole-brain map activated by both objects within and beyond and corresponding functional atlases (the pFs and LO from Zhen et al., 2015; the SPL and M1 from Fan et al., 2016). To achieve ROIs of similar sizes, we applied varying thresholds to each cortical area: for the pFs and LO, the atlases were thresholded at 55% and 90%, resulting in 266 voxels in the lpFs, 427 in the rpFs, 254 in the lLO and 347 in the rLO; for the SPL and M1, the atlases were thresholded at 78% and 94%, resulting in 661 voxels in the lSPL, 455 in the rSPL, 378 in the lM1, and 449 in the rM1. In the subsequent analysis, homologous areas spanning both cortical hemispheres were merged.”

(4) Discussion and theoretical implications. The authors discuss that the MRI results are consistent with the idea we only represent affordances within body size range. But the interpretation of the behavioural correlation matrices was that there was this similarity also for objects larger than body size, but forming a distinct cluster. I therefore found the interpretation of the MRI data inconsistent with the behavioural findings.

R4: We speculated that the similarity in action perception among objects beyond the body size range may be due to these objects being similarly conceptualized as‘environment’, in contrast to the objects within the body size range, which are categorized differently, namely as the ‘objects for the animal.’ Accordingly, in cortical regions involved in object processing, objects conceptualized as‘environment’ unlikely showed the congruency effect, distinct from objects within the body size range. We have explained this point in the revised manuscript:

(p 17, ln 370) “…which resonates the embodied influence on the formation of abstract concepts (e.g., Barsalou, 1999; Lakoff & Johnson, 1980) of objects and environment. Consistently, our fMRI data did not show the congruency effect for objects beyond the body size range, distinct from objects within this range, suggesting a categorization influenced by objects’ relative size to the human body.”

(5) In the discussion, the authors outline how this work is consistent with the idea that conceptual and linguistic knowledge is grounded in sensorimotor systems. But then reference Barsalou. My understanding of Barsalou is the proposition of a connectionist architecture for conceptual representation. I did not think sensorimotor representation was privileged, but rather that all information communicates with all other to constitute a concept.

R5: We are sorry for the confusion. We do not intend to argue that the sensorimotor representation is privileged. Instead, we would like to simply emphasize their engagement in concept. According to our understanding, Barsalou’s Perceptual Symbol Theory proposes that grounded concepts include sensorimotor information, and conceptual knowledge is grounded in the same neural system that supports action (Barsalou, 1999). This is consistent with our proposal that the affordance boundary locked to an animal’s sensorimotor capacity might give rise to a conceptual-ish representation of object-ness specific to the very animal. We have clarified this point in the introduction and discussion on the conceptual knowledge and sensorimotor information:

In the introduction (p 2, ln 59) “…, and the body may serve as a metric that facilitates meaningful engagement with the environment by differentiating objects that are accessible for interactions from those not. Further, grounded cognition theory (see Barsalou, 2008 for a review) suggests that the outputs of such differentiation might transcend sensorimotor processes and integrate into supramodal concepts and language. From this perspective, we proposed two hypotheses...”

In the discussion (p 18, ln 392) “Indeed, it has been proposed that conceptual knowledge is grounded in the same neural system that supports action (Barsalou, 1999; Glenberg et al., 2013; Wilson & Golonka, 2013), thereby suggesting that sensorimotor information, along with other modal inputs, may be embedded in language (e.g., Casasanto, 2011; Glenberg & Gallese, 2012; Stanfield & Zwaan, 2001), as the grounded theory proposed (see Barsalou, 2008 for a review).”

(6) More generally, I believe that the impact and implications of this study would be clearer for the reader if the authors could properly entertain an alternative concerning how objects may be represented. Of course, the authors were going to demonstrate that objects more similar in size afforded more similar actions. It was impossible that Ps would ever have responded that aeroplanes afford grasping and balls afford sitting, for instance. What do the authors now believe about object representation that they did not believe before they conducted the study? Which accounts of object representation are now less likely?

R6: We thank the reviewer for this suggestion. The theoretical motivation of the present study is to explore whether, for continuous action-related physical features (such as object size relative to the agents), affordance perception introduces discontinuity and qualitative dissociation, i.e., to allow the sensorimotor input to be assigned into discrete states/kinds, as representations envisioned by the computationalists; alternatively, whether the activity may directly mirror the input, free from discretization/categorization/abstraction, as proposed by the Replacement proposal of some embodied theories on cognition.

By addressing this debate, we hoped to shed light on the nature of representation in, and resulted from, the vision-for-action processing. Our finding of affordance discontinuity suggests that sensorimotor input undergoes discretization implied in the computationalism idea of representation. Further, not contradictory to the claims of the embodied theories, these representations do shape processes out of the sensorimotor domain, but after discretization.

We have now explained our hypotheses and alternatives explicitly in the revised introduction and discussion:

In the introduction (p 2, ln 45) “However, the question of how object perception is influenced by the relative size of objects in relation to the human body remains open. Specifically, it is unclear whether this relative size simply acts as a continuous variable for locomotion reference, or if it affects differentiating and organizing object representation based on their ensued affordances.”

In the discussion (p 14, ln 295) “One long-lasting debate on affordance centers on the distinction between representational and direct perception of affordance. An outstanding theme shared by many embodied theories of cognition is the replacement hypothesis (e.g., Van Gelder, 1998), which challenges the necessity of representation as posited by computationalism’s cognitive theories (e.g., Fodor, 1975). This hypothesis suggests that input is discretized/categorized and subjected to abstraction or symbolization, creating discrete stand-ins for the input (e.g., representations/states). Such representationalization would lead to a categorization between the affordable (the objects) and those beyond affordance (the environment), in contrast to the perspective offered by embodied theories. The present study probed this ‘representationalization’ of affordance by examining whether affordance perception introduces discontinuity and qualitative dissociation in response to continuous action-related physical features (such as object size relative to the agents), which allows sensorimotor input to be assigned into discrete states/kinds, in line with the representation-based view under the constraints of body size. Alternatively, it assessed whether activity directly mirrors the input, free from discretization/categorization/abstraction, in line with the representation-free view.

First, our study found evidence demonstrating discretization in affordance perception. Then, through the body imagination experiment, we provided causal evidence suggesting that this discretization originates from sensorimotor interactions with objects rather than amodal sources, such as abstract object concepts independent of agent motor capability. Finally, we demonstrated the supramodality of this embodied discontinuity by leveraging the recent advances in AI. We showed that the discretization in affordance perception is supramodally accessible to disembodied agents such as large language models (LLMs), which lack sensorimotor input but can access linguistic materials built upon discretized representations. These results collectively suggest that sensorimotor input undergoes discretization, as implied in the computationalism’s idea of representation. Note that, these results are not contradictory to the claim of the embodied theories, as these representations do shape processes beyond the sensorimotor domain but after discretization.

This observed boundary in affordance perception extends the understanding of the discontinuity in perception in response to the continuity of physical inputs (Harnad, 1987; Young et al., 1997).”

**Reviewer #1 (Recommendations For The Authors):**
a) I would recommend providing further justification for why 100-200 cm were used as the cut-offs reflecting acceptable imagined body size. Were these decisions preregistered anywhere? If so, please state.

Ra: Please see R1.

b) I would encourage the authors to call the MRI a small pilot study throughout, including in the abstract.

Rb: We completely agree and have indicated the preliminary nature of this study in the revised version:

(p 11, ln 236) “To test this speculation, we ran an fMRI experiment with a small number of participants to preliminarily investigate the neural basis of the affordance boundary in the brain by measuring neural activity in the dorsal and ventral visual streams when participants were instructed to evaluate whether an action was affordable by an object (Fig. 3a).”

c) Please provide much further justification of ROI selection, why these thresholds were chosen, and therefore why they are different across regions.

Rc: Please see R3.

d) Further elucidation in the discussion would help the reader interpret the MRI data, which should always be interpreted also in light of the behavioural findings.

Rd: Please see R4.

e) The authors may wish to outline precisely what they claim concerning the nature of conceptual/linguistic representation. Is sensorimotor information privileged or just part of the distributed representation of concepts?

Re: This is a great point. For details of corresponding revision, please see R5.

f) There are some nods to alternative manners in which we plausibly represent objects (e.g. about what the imagination study tells us) but I think this theoretical progression should be more prominent.

Rf: We thank the reviewer for this suggestion. For details of corresponding revision, please see R6.

**Reviewer #2 (Public Review):**
(1) A limitation of the tests of LLMs may be that it is not always known what kinds of training material was used to build these models, leading to a possible "black box" problem. Further, presuming that those models are largely trained on previous human-written material, it may not necessarily be theoretically telling that the "judgments" of these models about action-object pairs show human-like discontinuities. Indeed, verbal descriptions of actions are very likely to mainly refer to typical human behaviour, and so the finding that these models demonstrate an affordance discontinuity may simply reflect those statistics, rather than evidence that affordance boundaries can arise independently even without "organism-environment interactions" as the authors claim here.

R1: We agree that how LLMs work is a “black box”, and thus it is speculative to assume them to possess any human-like ability, because, as the reviewer pointed out, “these models demonstrate an affordance discontinuity may simply reflect those statistics.” Indeed, our manuscript has expressed a similar idea: “We speculated that ChatGPT models may have formed the affordance boundary through a human prism ingrained within its linguistic training corpus. (p 16 ln 338)”. That is, we did not intend to claim that such information is sufficient to replace sensorimotor-based interaction, or to restore human-level capability, for which we indeed speculated that embodied interaction is necessary. In the revised manuscript, we have clarified our stand that the mechanism generating the observed affordance boundary in LLMs might be different from that in human cognition, and urged future studies to explore this possibility:

(p 18, ln 413) “…, as well as alignment methods used in fine-tuning the model (Ouyang et al., 2022). Nevertheless, caution should be taken when interpreting the capabilities of LLMs like ChatGPT, which are often considered “black boxes.” That is, our observation indicates that some degree of sensorimotor information is embedded within human language materials presumably through linguistic statistics, but it is not sufficient to assert that LLMs have developed a human-like ability to represent affordances. Furthermore, such information alone may be insufficient for LLMs to mimic the characteristics of the affordance perception in biological intelligence. Future studies are needed to elucidate such limitation.”

Indeed, because of this potential dissociation, our LLM study might bear novel implications for the development of AI agents. We elaborated on them in the revised discussion on LLMs:

(p 19, ln 427) “…, represents a crucial human cognitive achievement that remains elusive for AI systems. Traditional AI (i.e., task-specific AI) has been confined with narrowly defined tasks, with substantial limitations in adaptability and autonomy. Accordingly, these systems have served primarily as tools for humans to achieve specific outcomes, rather than as autonomous agents capable of independently formulating goals and translating them into actionable plans. In recent years, significant efforts have been directed towards evolving traditional AI into more agent-like entities, especially in domains like navigation, object manipulation, and other interactions with the physical world. Despite these advancements, the capabilities of AI still fall behind human-level intelligence. On the other hand, embodied cognition theories suggest that sensorimotor interactions with the environment are foundational for various cognitive domains. From this point of view, endowing AI with human-level abilities in physical agent-environment interactions might provide an unreplaceable missing piece for achieving Artificial General Intelligence (AGI). This development would significantly facilitate AI’s role in robotics, particularly in actions essential for survival and goal accomplishment, a promising direction for the next breakthrough in AI (Gupta et al., 2021; Smith & Gasser, 2005).

However, equipping a disembodied AI with the ability for embodied interaction planning within a specific environment remains a complex challenge. By testing the potential representationalization of action possibilities (affordances) in both humans and LLMs, the present study suggests a new approach to enhancing AI’s interaction ability with the environment. For instance, our finding of supramodal affordance representation may indicate a possible pathway for disembodied LLMs to engage in embodied physical interactions with their surroundings. From an optimistic view, these results suggest that LLM-based agents, if appropriately designed, may leverage affordance representations embedded in language to interact with the physical world. Indeed, by clarifying and aligning such representations with the physical constitutes of LLM-based agents, and even by explicitly constructing an agent-specific object space, we may foster the sensorimotor interaction abilities of LLM-based agents. This progression could lead to achieving animal-level interaction abilities with the world, potentially sparking new developments in the field of embodied cognition theories.”

(2) The authors include a clever manipulation in which participants are asked to judge action-object pairs, having first adopted the imagined size of either a cat or an elephant, showing that the discontinuity in similarity judgments effectively moved to a new boundary closer to the imagined scale than the veridical human scale. The dynamic nature of the discontinuity suggests a different interpretation of the authors' main findings. It may be that action affordance is not a dimension that stably characterises the long-term representation of object kinds, as suggested by the authors' interpretation of their brain findings, for example. Rather these may be computed more dynamically, "on the fly" in response to direct questions (as here) or perhaps during actual action behaviours with objects in the real world.

R2: We thank the reviewer for pointing out the dynamic nature of affordance perception in our study. This feature indeed reinforced our attribution of the boundary into an affordance-based process instead of a conceptual or semantic process, the latter of which would predict the action possibilities being a fixed belief about the objects, instead of being dynamically determined according to the feature of the agent-object dyads. In addition, this dynamic does not contradict with our interpretation of the observed boundary in affordance perception. With this observation, we speculated that continuous input was abstracted or representationalized into discontinued categories, and the boundary between these categories was drawn according to the motor capacity of the agent. The finding of the boundary adapting to manipulation on body schema suggests that the abstraction/representationalization dynamically updates according to the current belief of motor capacity and body schema of the animal. In addition, we agree that future studies are needed to examine the dynamics of the abstraction/representationalization of affordance, probably by investigating the evolvement of affordance representation during ongoing actual interactions with novel objects or manipulated motor capability. These points are now addressed in the revision:

(p 17, ln 380) “Therefore, this finding suggests that the affordance boundary is cognitively penetrable, arguing against the directness of affordance perception (e.g., Gibson, 1979; Greeno, 1994; Prindle et al., 1980) or the exclusive sensorimotor origin of affordances (e.g., Gallagher, 2017; Thompson, 2010; Hutto & Myin, 2012; Chemero, 2013). Further, this finding that the boundary adapted to manipulation on body schema suggests that the abstraction/representationalization may be dynamically updated in response to the current motor capacity and body schema of the agent, suggesting that the affordance-based process is probably determined dynamically by the nature of the agent-object dyads, rather than being a fixed belief about objects. Future studies could explore the dynamics of affordance representationalization, probably by investigating how affordance representations evolve during active interactions with novel objects or under conditions of altered motor capabilities. Finally, our findings also suggest that disembodied conceptual knowledge pertinent to action likely modulates affordance perception.”

**Reviewer #2 (Recommendations For The Authors):**
a) As described, I think the authors could improve their discussion of the LLM work and consider more deeply possible different interpretations of their findings with those models. Are they really providing an independent data point about how objects may be represented, or instead is this a different, indirect way of asking humans the same questions (given the way in which these models are trained)?

Ra: Please see R1.

b) Some of the decisions behind the design of the fMRI experiment, and some of the logic of its interpretation, could be made clearer. Why those four objects per se? What kinds of confounds, such as familiarity, or the range of possible relevant actions per object, might need to be considered? Is there the possibility that relative performance on the in-scanner behavioural task may be in part responsible for the findings? Why were those specific regions of interest chosen and not others? The authors find that the dorsal and ventral regions make a univariate distinction between congruent and incongruent trials, but only for human-scale objects, but it was not clear from the framework that the authors adopted why that distinction should go in that direction (e.g. congruent > incongruent) nor why there shouldn't also be a distinction for the "beyond" objects? Finally, might some of these brain questions better be approached with an RSA or similar approach, as that would seem to better map onto the behavioural studies?

Rb: We thank the reviewer for the detailed suggestions.

Regarding the fMRI study, we have provided further justification on its rationale in the revised manuscript:

(p 11, ln 231) “The distinct categories of reported affordances demarcated by the boundary imply that the objects on either side of the boundary may be represented differently in the brain. We thus speculated that the observed behavioral discontinuity is likely underpinned by distinct neural activities, which give rise to these discrete ‘representations’ separated by the boundary.”

The objects used in the fMRI study were selected by taking into account the objective of the fMRI study, which was to provide the neural basis for the affordance discontinuity found in behaviour experiments. In other words, the fMRI study is not an exploratory experiment, but a validation experiment. To this end, we deliberately selected a small range of common objects to ensure that participants were sufficiently familiar with them, as confirmed through their oral reports. Furthermore, to ensure a fair comparison between the two categories of objects in terms of action possibility range, we predetermined an equal number of congruent and incongruent actions for each category. This arrangement was intended to eliminate any bias that might arise from different amount of action choices associated with each category. Therefore, the present object and action sets in the fMRI study, which were based on the behavior experiments, are sufficient for its purpose.

Regarding the possibility that the performance of the in-scanner behavioural task may be in part responsible for the findings, we analysed participants’ performance. Not surprisingly, participants demonstrated high consistency and accuracy in their responses:

*MeanCongruent_ObjectWithin* = 0.991, SD = 0.018;

*MeanIncongruent_ObjectWithin* = 0.996, SD = 0.007;

*MeanCongruent_ObjectBeyond* = 0.996, SD = 0.004;

*MeanIncongruent_ObjectBeyond* = 0.998, SD = 0.002

in all conditions, suggesting constant active engagement with the task. Thus, the inscanner behaviour unlikely resulted in the lack of congruency effect for the ‘beyond’ objects observed in the brain.

Regarding the selection of ROIs, our decision to focus on these specific sensory and motor regions was based on existing literature highlighting their distinct contribution to affordance perception (Borghi, 2005; Sakreida et al., 2016). The pFs was chosen for its role in object identification and classification, while the SPL was chosen for its involvement in object manipulation. Additionally, the primary motor cortex (M1) is known to be engaged in affordance processing (e.g., McDannald et al., 2018), which was included to investigate the affordance congruency effect during the motor execution stage of the sense-think-act pathway. These considerations are detailed in the revised manuscript:

(p 14, ln 280) “In addition to the pFs and SPL, we also examined the congruency effect in the lateral occipital cortex (LO), which is involved in object representation (e.g., Grill-Spector et al., 2000; Konkle & Caramazza, 2013) and provides inputs to both the pFs and SPL (Hebart et al., 2018). Meanwhile, the primary motor cortex (M1), which receives inputs from the dorsal stream (Vainio & Ellis, 2020), is involved in affordance processing (e.g., McDannald et al., 2018) and action executions(Binkofski et al., 2002).”

(p 29, ln 684) “We chose the pFs, LO, SPL, and M1 as ROIs based on existing literature highlighting their distinct contributions to affordance perception (Borghi,2005; Sakreida et al., 2016).”

Regarding the congruency effect, in our study, we followed the established fMRI research paradigm of employing the congruent effect as a measure of affordance processing (e.g., Kourtis et al., 2018), and the rationale behind the directionality of the distinction in our framework (congruent > incongruent) is grounded in the concept of affordance, in which the mere perception of a graspable object facilitates motor responses that are congruent with certain qualities of the object (e.g., Ellis & Tucker, 2000). From the interaction of congruency by object type, we observed only congruency effect for objects within rather than objects beyond. We speculate that the objects beyond the affordance boundary is generally beyond the motor capacities of the very animal, being too large for the animal to manipulate, thus no congruency effect was found. We have added these clarifications in the revised manuscript:

(p 11, ln 244) “The congruency effect, derived from the contrast of Congruent versus Incongruent conditions, is a well-established measure of affordance processing(e.g., Kourtis et al., 2018).”

(p 16, ln 340) “In contrast, objects larger than that range typically surpass the animal’s motor capabilities, rendering them too cumbersome for effective manipulation. Consequently, these larger objects are less likely to be considered as typical targets for manipulation by the animal, as opposed to the smaller objects. That is, they are perceived not as the “objects” in the animal’s eye, but as part of the background environment, due to their impracticality for direct interactions.”

Regarding the RSA analysis, we agree with the reviewer that RSA may offer a more direct comparison with similarities among objects. However, our primary objective in this fMRI study was to explore the neural basis of the affordance boundary observed in the behavioural study, rather than explaining the similarities in neural responses between different objects. For this reason, we did not conduct RSA analysis.

c) Page 4 Re statistical evaluation of the discontinuity in judgments, the authors might consider a Bayesian approach, which would be stronger than using "all ps > 0.05" to argue that within-boundary similarities are consistent and high.

Rc: We thank the reviewer for the suggestion on the Bayesian approach for significance tests, which has been now added in the revised manuscript:

In the results (p 4, ln 105) “This trough suggested an affordance boundary between size rank 4 and 5, while affordance similarities between neighboring ranks remained high (rs > 0.45) and did not significantly differ from each other (ps > 0.05, all *BF*bb10 < 10) on either side of the boundary (Fig. 1d, left panel, green lines).”

In the methods (p 25, ln 597) “Pearson and Filon’s (1898) Z, implemented in R package “cocor” (Diedenhofen & Musch, 2015) was used to evaluate the significance of these similarities (alpha level = .05, one-tail test). For significance tests, Bayesian statistical analyses were conducted using the web version of the “bayesplay” R package (Colling, 2021). Specifically, the data (likelihood) model was specified as a normal distribution, where the correlation coefficients were transformed to Fisher’s z. The null hypothesis was specified as a standard normal distribution centred at zero. Conversely, the alternative hypothesis was specified as a normal distribution centred at 2. Bayes factors (BF10) were calculated and interpreted using the classification scheme suggested by Wagenmakers et al. (2011), wherein a Bayes factor greater than 10 is considered strong evidence for accepting H1 over H0.”

d) Page 4 One question I had about the big objects is whether their internal similarity and dissimilarity to smaller objects, might largely arise if most of the answers about actions for those larger objects are just "no"? This depends on the set of possible actions that were considered: the authors chose 14 from a previous study but did not describe these further or consider possible strengths/limitations of this selection. This is a very important point that needs addressing - to what extent are these findings "fragile" in that they relate only to that specific selection of 14 action kinds?

Rd: The action judgements for objects beyond body size were not mostly “no”; in fact, there was no significant difference between average action possibilities related to objects beyond (25%) and within (26%). Rather, the dissimilarity between objects within and those beyond likely arose from the difference in most-plausible action set they related. For example, the top three actions related to objects within are “grasp”, “hold” and “throw”, while those related to objects beyond are “sit”, “lift” and “stand”, as stated in our original manuscript: “A further analysis on the affordances separated by the boundary revealed that objects within human body size range were primarily subjected to hand-related actions such as grasping, holding and throwing. These affordances typically involve object manipulation with humans’ effectors. In contrast, objects beyond the size range of human body predominantly afforded actions such as sitting and standing, which typically require locomotion or posture change of the whole body around or within the objects (p 11 ln 229)”.

Regarding the validity of action selection, the selection of the objects and affordances in this study was guided by two key criteria. First, the objects were selected from the dataset published in Konkle and Oliva's study (2011), which systematically investigates the effect of object size on object recognition. Therefore, the range of object sizes, from 14 cm to 7,618 cm, is well-calibrated and represents a typical array of object sizes found in the real world. Second, the actions were selected to cover a wide range of daily humans-objects/environments interactions, from singlepoint movements (e.g., hand, foot) to whole-body movements (e.g., lying, standing), based on the kinetics human action video dataset (Kay et al., 2017). Thus, this set of objects and actions is a sufficiently representative of typic human experiences. In revision, we have clarified these two criteria in the methods section:

(p 22, ln 517) “The full list of objects, their diagonal size, and size rankings were provided in Supplementary Table S6. The objects were selected from the dataset in Konkle and Oliva’s study (2011) to cover typic object sizes in the world (ranging from 14 cm to 7,618 cm), and actions related to these objects were selected to span a spectrum of daily humans-objects/environments interactions, from single-point movements (e.g., hand, foot) to whole-body movements (e.g., lying, standing), based on the Kinetics Human Action Video Dataset (Kay et al., 2017).”

Having said this, we agree with reviewer that a larger set of objects and actions will facilitate finer localization of the representational discontinuity, which can be addressed in future studies

(p 16, ln 344): “…, due to their impracticality for direct interactions. Future studies should incorporate a broader range of objects and a more comprehensive set of affordances for finer delineation of the representational discontinuity between objects and the environment.”

e) Page 12 "no region showed the congruency effect for objects beyond the body size" in a whole brain analysis. What about a similar analysis for the humanscale objects? We must also keep in mind that with N=12 there may be relatively little power to detect such effects at the random-effects level, so this null finding may not be very informative.

Re: We thank the reviewer for this advice. The whole brain analysis on the congruency effect for human-scale objects (objects within) has now been included in the supplementary materials (please see Author response figure 1d (New Supplementary Fig. S4d) and Author response table 1 (New Supplementary Table S5) below)Author response table 1.

**Author response table 1. sa4table1:** Cortical regions showing significant congruency effect (congruent versus incongruent) for objects within, whole-brain analysis (R = right hemisphere, L = left hemisphere; Z > 2.3, p = 0.05, cluster corrected).

Cluster	Region	Number ofvoxels in region	MNI coordinates	Peak Z value
x	y	z
(OW_Congruent – OW_Incongruent) > (OB_Congruent – OB_Incongruent)
1	L Middle Occipital Gyrus	831	22	–94	10	4.25
R Middle Occipital Gyrus	187
L Fusiform Gyrus	113
R Fusiform Gyrus	376
L Inferior Occipital Gyrus	293
R Inferior Occipital Gyrus	276
L Lingual Gyrus	215
R Lingual Gyrus	345
L Superior Occipital Gyrus	123
2	R Supplementary Motor Area	383	14	14	60	3.39
3	R Superior Parietal Lobule	191	36	–62	56	3.18
R Inferior Parietal Lobule	114
4	R Insula	175	32	18	8	3.41
(OB_Congruent – OB_Incongruent) > (OW_Congruent – OW_Incongruent)
-	No significant cluster	-	-	-	-	-

**Author response image 1. sa4fig1:** Significant brain activations of different contrasts in the whole-brain level analysis. (a) the effect of object type, positive values (warm color) indicated higher activation for objects within than objects beyond and negative values (cold color) indicated the opposite. (b) the effect of congruency, positive values indicated higher activation in congruent than incongruent condition. (c) the effect of interaction between object type and congruency, positive values indicated the larger congruency effect for objects within than beyond. (d) the congruency effect for objects within. All contrasts were corrected with cluster-level correction at p < .05. The detailed cluster-level results for each contrast map can be found in Supplementary Table S2 to S5.

Regarding the power of the fMRI study, we would like to clarify that, the critical test of this fMRI study is the two-way interaction of congruency effect by object size instead of the (null) congruency effect for the object beyond. Having said this, we agree that the sample size is small which might lead to lack of power in the fMRI study. In the revision we have now acknowledged this issue explicitly:

(p 16, ln 354) “…supporting the idea that affordance is typically represented only for objects within the body size range. While it is acknowledged that the sample size of the fMRI study was small (12 participants), necessitating cautious interpretation of its results, the observed neural-level affordance discontinuity is notable. That is, qualitative differences in neural activity between objects within the affordance boundary and those beyond replicated our behavior findings. This convergent evidence reinforced our claim that objects were discretized into two broad categories along the continuous size axis, with affordance only being manifested for objects within the boundary.”

f) Page 14 [the fMRI findings] "suggest that affordance perception likely requires perceptual processing and is not necessarily reflected in motor execution". This seems a large leap to make from a relatively basic experiment that tests only a small set of (arbitrarily chosen) objects and actions. It's important to keep in mind too that none of the studies here actually asked participants to interact with objects; that objects were shown as 2D images; and that the differences between real-world sizes of objects were greatly condensed by the way they are scaled for presentation on a computer screen (and such scaling is probably greater for the larger-than-human objects).

Rf: The action-congruency judgement task is widely used in the studies of affordance processing (e.g., Kourtis et al., 2018; Peelen & Caramazza, 2012), so does the practice of not including actual interaction with the objects and using 2D instead of 3D objects(e.g., Peelen & Caramazza, 2012; Matić et al., 2020). However, we are aware that alternative practice exists in the field and we agree that it would be interesting for future studies to test whether actual interactions and 3D objects presentation may bring any change on the affordance boundary observed in our study.

Our inference “affordance perception likely requires perceptual processing and is not necessarily reflected in motor execution” was based on the fMRI finding that the congruency effect only in cortical regions proposedly engaged in perceptual processing, but not in the M1 which is associated with motor execution. This significant two-way interaction pointed to a possibility that affordance processing may not necessarily manifest in motor execution.

We acknowledge the scaling issue inherent in all laboratory experiments, but we doubt that it significantly influenced our results. In fact, it is a common practice in studies on object size to present objects of different physical sizes as constantly sized images on a screen (e.g., Konkle & Oliva, 2012; Huang et al., 2022). Moreover, scaling does not change the smoothness of object sizes, whereas the affordance boundary represents a singularity point that disrupts this smoothness. Finally, regarding the limited variety of objects and actions, please see Rd.

g) Page 15 Why are larger objects "less interesting"? They have important implications for navigation, for example?

Rg: We are sorry for the confusion. Our intention was to express that objects beyond the affordance boundary are generally beyond motor capacities of the animal in question. As such, compared to smaller objects within the environment, these larger objects may not typically be considered as potential targets for manipulation. We have now corrected the wording in the revised text:

(p 16, ln 340) “In contrast, objects larger than that range typically surpass the animal’s motor capabilities, rendering them too cumbersome for effective manipulation. Consequently, these larger objects are less likely to be considered as typical targets for manipulation by the animal, as opposed to smaller objects in the environment. That is, they are perceived not as the “objects” in the animal’s eye, but as part of the background environment, due to their impracticality for direct interactions.”

h) Page 15 At several places I wondered whether the authors were arguing against a straw man. E.g. "existing psychological studies...define objects in a disembodied manner..." but no citations are given on this point, nor do the authors describe previous theoretical positions that would make a strong counter-claim to the one advocated here.

Rh: We are sorry for not presenting our argument clearly. Previous studies often define the object space based on object features alone, such as absolute size or function, without reference to the knowledge and the abilities of the agent (e.g., de Beeck et al., 2008; Konkle & Oliva, 2011). This perspective overlooks the importance of the features of the animal-object pairs. Gibson (1979) highlighted that an object’s affordance, which includes all action possibilities it offers to an animal, is determined by the object’s size relative to the animal’s size, rather than its real-world size. Under this embodied view, we argue that the object space is better defined by the features of the agent-object system, and this is the primary assumption and motivation of the present study. We have now clarified this point and added the references in the revision:

(p 2, ln 35) “A contemporary interpretation of this statement is the embodied theory of cognition (e.g., Chemero, 2013; Gallagher, 2017; Gibbs, 2005; Wilson, 2002; Varela et al., 2017), which, diverging from the belief that size and shape are inherent object features (e.g., de Beeck et al., 2008; Konkle & Oliva, 2011), posits that human body scale (e.g., size) constrains the perception of objects and the generation of motor responses.”

(p 17, ln 365) “Existing psychological studies, especially in the field of vision, define objects in a disembodied manner, primarily relying on their physical properties such as shape (e.g., de Beeck et al., 2008) and absolute size (e.g., Konkle & Oliva, 2011).”

**Reviewer #3 (Public Review):**
(1) Even after several readings, it is not entirely clear to me what the authors are proposing and to what extent the conducted work actually speaks to this. In the introduction, the authors write that they seek to test if body size serves not merely as a reference for object manipulation but also "plays a pivotal role in shaping the representation of objects." This motivation seems rather vague motivation and it is not clear to me how it could be falsified.Similarly, in the discussion, the authors write that large objects do not receive "proper affordance representation," and are "not the range of objects with which the animal is intrinsically inclined to interact, but probably considered a less interesting component of the environment." This statement seems similarly vague and completely beyond the collected data, which did not assess object discriminability or motivational values.Overall, the lack of theoretical precision makes it difficult to judge the appropriateness of the approaches and the persuasiveness of the obtained results. This is partly due to the fact that the authors do not spell out all of their theoretical assumptions in the introduction but insert new "speculations" to motivate the corresponding parts of the results section. I would strongly suggest clarifying the theoretical rationale and explaining in more detail how the chosen experiments allow them to test falsifiable predictions.

R1: We are sorry for the confusion about the theoretical motivation and rationale. Our motivation is on the long-lasting debate regarding the representation versus direct perception of affordance. That is, we tested whether object affordance would simply covary with its continuous constraints such as object size, in line with the representation-free view, or, whether affordance would be ‘representationalized’, in line with the representation-based view, under the constrain of body size. In revision, we have clarified the motivation and its relation to our approach:

In the introduction (p 2, ln 45): “However, the question of how object perception is influenced by the relative size of objects in relation to the human body remains open. Specifically, it is unclear whether this relative size simply acts as a continuous variable for locomotion reference, or if it affects differentiating and organizing object representations based on their ensued affordances.”

In the discussion (p 14, ln 295): “One long-lasting debate on affordance centers on the distinction between representational and direct perception of affordance. An outstanding theme shared by many embodied theories of cognition is the replacement hypothesis (e.g., Van Gelder, 1998), which challenges the necessity of representation as posited by computationalism’s cognitive theories (e.g., Fodor, 1975). This hypothesis suggests that input is discretized/categorized and subjected to abstraction or symbolization, creating discrete stand-ins for the input (e.g., representations/states). Such representationalization would lead to a categorization between the affordable (the objects) and those beyond affordance (the environment). Accordingly, computational theories propose the emergence of affordance perception, in contrast to the perspective offered by embodied theories. The present study probed this ‘representationalization’ of affordance by examining whether affordance perception introduces discontinuity and qualitative dissociation in response to continuous action-related physical features (such as object size relative to the agents), which allows sensorimotor input to be assigned into discrete states/kinds, in line with the representation-based view under the constraints of body size. Alternatively, it assessed whether activity directly mirrors the input, free from discretization/categorization/abstraction, in line with the representation-free view.

First, our study found evidence demonstrating discretization in affordance perception. Then, through the body imagination experiment, we provided causal evidence suggesting that this discretization originates from sensorimotor interactions with objects rather than amodal sources, such as abstract object concepts independent of agent motor capability. Finally, we demonstrated the supramodality of this embodied discontinuity by leveraging the recent advances in AI. We showed that the discretization in affordance perception is supramodally accessible to disembodied agents such as large language models (LLMs), which lack sensorimotor input but can access linguistic materials built upon discretized representations. These results collectively suggest that sensorimotor input undergoes discretization, as implied in the computationalism’s idea of representation. Note that, these results are not contradictory to the claim of the embodied theories, as these representations do shape processes beyond the sensorimotor domain but after discretization.

The observed boundary in affordance perception extends the understanding of the discontinuity in perception in response to the continuity of physical inputs (Harnad, 1987; Young et al., 1997).”

We are also sorry for the confusion about the expression “proper affordance representation”. We intended to express that the neural responses to objects beyond the boundary in the whole brain failed to reflect affordance congruency, and therefore did not show evidence of affordance processing. We have clarified this expression in the revised manuscript:

(p 12, ln 265) “Taken together, the affordance boundary not only separated the objects into two categories based on their relative size to human body, but also delineated the range of objects that evoked neural representations associated with affordance processing.”

Finally, we agree with the reviewer that the expressions, such as “not…inclined to interact” and “probably considered a less interesting component of the environment”, may be misleading. Rather, we intended to express that the objects beyond the affordance boundary is generally beyond the motor capacities of the very animal, being too large for the very animal to manipulated, as comparing to the smaller objects in the environment, may not be a typical target object for manipulation for the animal. We have revised these expressions in the manuscript and clarified their speculative nature:

(p 16, ln 340) “In contrast, objects larger than that range typically surpass the animal’s motor capabilities, rendering them too cumbersome for effective manipulation. Consequently, these larger objects are less likely to be considered as typical targets for manipulation by the animal, as opposed to the smaller objects. That is, they are perceived not as the “objects” in the animal’s eye, but as part of the background environment, due to their impracticality for direct interactions.”

(2) The authors used only a very small set of objects and affordances in their study and they do not describe in sufficient detail how these stimuli were selected. This renders the results rather exploratory and clearly limits their potential to discover general principles of human perception. Much larger sets of objects and affordances and explicit data-driven approaches for their selection would provide a far more convincing approach and allow the authors to rule out that their results are just a consequence of the selected set of objects and actions.

R2: The selection of the objects and affordances in this study was guided by two key criteria. First, the objects were selected from the dataset published in Konkle and Oliva's study (2011), which systematically investigates the effect of object size on object recognition. Therefore, the range of object sizes, from 14 cm to 7,618 cm, is well-calibrated and represents a typical array of object sizes found in the real world. Second, the actions were selected to cover a wide range of daily humans objects/environments interactions, from single-point movements (e.g., hand, foot) to whole-body movements (e.g., lying, standing), based on the kinetics human action video dataset (Kay et al., 2017). Thus, this set of objects and actions is a sufficiently representative of typic human experiences. In revision, we have clarified these two criteria in the methods section:

(p 22, ln 517) “The full list of objects, their diagonal sizes, and size rankings were provided in Supplementary Table S6. The objects were selected from the dataset in Konkle and Oliva’s study (2011) to cover typic object sizes in the world (ranging from 14 cm to 7,618 cm), and actions related to these objects were selected to span a spectrum of daily humans-objects/environments interactions, from single-point movements (e.g., hand, foot) to whole-body movements (e.g., lying, standing), based on the Kinetics Human Action Video Dataset (Kay et al.,2017).”

Having said this, we agree with reviewer that a larger set of objects and actions will facilitate finer localization of the representational discontinuity, which can be addressed in future studies

(p 16, ln 344): “…, due to their impracticality for direct interactions. Future studies should incorporate a broader range of objects and a more comprehensive set of affordances for finer delineation of the representational discontinuity between objects and the environment.”

(3) Relatedly, the authors could be more thorough in ruling out potential alternative explanations. Object size likely correlates with other variables that could shape human similarity judgments and the estimated boundary is quite broad (depending on the method, either between 80 and 150 cm or between 105 to 130 cm). More precise estimates of the boundary and more rigorous tests of alternative explanations would add a lot to strengthen the authors' interpretation.

R3: We agree with the reviewer that correlation analyses alone cannot rule out alternative explanations, as any variable co-varying with object sizes might also affect affordance perception. Therefore, our study experimentally manipulated the imagined body sizes, while keeping other variable constant across conditions. This approach provided evidence of a causal connection between body size and affordance perception, effectively ruling out alternative explanations. In revision, the rationale of experimentally manipulation of imagined body sizes has been clarified

(p 7, ln 152): “One may argue that the location of the affordance boundary coincidentally fell within the range of human body size, rather than being directly influenced by it. To rule out this possibility, we directly manipulated participants’ body schema, referring to an experiential and dynamic functioning of the living body within its environment (Merleau-Ponty & Smith, 1962). This allowed us to examine whether the affordance boundary would shift in response to changes in the imagined body size. This experimental approach was able to establish a causal link between body size and affordance boundary, as other potential factors remained constant. Specifically, we instructed a new group of participants to imagine themselves as small as a cat (typical diagonal size: 77cm, size rank 4, referred to as the “cat condition”), and another new group to envision themselves as large as an elephant (typical diagonal size: 577 cm, size rank 7, referred to as the “elephant condition”) throughout the task (Fig. 2a).”

Meanwhile, with correlational analysis, precise location of the boundary cannot help ruling out alternative explanation. However, we agree that future studies are needed to incorporate a broader range of objects and a more comprehensive set of affordances. For details, please see R2.

(4) Even though the division of the set of objects into two homogenous clusters appears defensible, based on visual inspection of the results, the authors should consider using more formal analysis to justify their interpretation of the data. A variety of metrics exist for cluster analysis (e.g., variation of information, silhouette values) and solutions are typically justified by convergent evidence across different metrics. I would recommend the authors consider using a more formal approach to their cluster definition using some of those metrics.

R4: We thank the reviewer for the suggestion. We performed three analyses on this point, all of which consistently indicated the division of objects into two distinct groups along the object size axis.

First, a hierarchical clustering analysis of the heatmaps revealed a two-maincluster structure, which is now detailed in the revised methods section (p 25, ln 589) “A hierarchical clustering analysis was performed, employing the seaborn clustermap method with Euclidean distance and Complete linkage (Waskom, 2021).”

Second, the similarity in affordances between neighbouring size ranks revealed the same two-main-cluster structure. In this analysis, each object was assigned a realworld size rank, and then Pearson’s correlation was calculated as the affordance similarity index for each pair of neighbouring size ranks to assess how similar the perceived affordances were between these ranks. Our results showed a clear trough in affordance similarity, with the lowest point approaching zero, while affordance similarities between neighbouring ranks on either side of the boundary remained high, confirming the observation that objects formed two groups based on affordance similarity.

Finally, we analysed silhouette values for this clustering analysis, where *a*_*i*_ represents the mean intra-cluster distance, and *b*_*i*_ represents the mean nearest-cluster distance for each data point i. The silhouette coefficient is calculated as (Rousseeuw, 1987):Si=bi−aimax(bi,ai)

The silhouette analysis revealed that the maximum silhouette value coefficient corresponded to a cluster number of two, further confirming the two-cluster structure (please see Author response table 2 below).

**Author response table 2. sa4table2:** The silhouette values of a k-means clustering when k (number of clusters) = 2 to 10.

Number of clusters	2	3	4	5	6	7	8	9	10
Average silhouette values	0.362	0.275	0.299	0.194	0.171	0.263	0.176	0.164	0.137

(5) While I appreciate the manipulation of imagined body size, as a way to solidify the link between body size and affordance perception, I find it unfortunate that this is implemented in a between-subjects design, as this clearly leaves open the possibility of pre-existing differences between groups. I certainly disagree with the authors' statement that their findings suggest "a causal link between body size and affordance perception."

R5: The between-subjects design in the imagination experiment was employed to prevent contamination between conditions. Specifically, after imagining oneself as a particular size, it can be challenging to immediately transition to envisioning a different body size. In addition, participating sequentially participate in two conditions that only differ in imagined body sizes may lead to undesirable response strategies, such as deliberately altering responses to the same objects in the different conditions. The reason of employing the between-subjects design is now clarified in the revised text (p 7, ln 161): “A between-subject design was adopted to minimize contamination between conditions. This manipulation was effective, as evidenced by the participants’ reported imagined heights in the cat condition being 42 cm (SD = 25.6) and 450 cm (SD = 426.8) in the elephant condition on average, respectively, when debriefed at the end of the task.”

Further, to address the concern that “pre-existing differences between groups” would generate this very result, we adhered to standard protocols such as random assignment of participants to different conditions (cat-size versus elephant-size). Moreover, experimentally manipulating one variable (i.e., body schema) to observe its effect on another variable (i.e., affordance boundary) is the standard method for establishing causal relationships between variables. We could not think of other better ways for this objective.

(6) The use of LLMs in the current study is not clearly motivated and I find it hard to understand what exactly the authors are trying to test through their inclusion. As noted above, I think that the authors should discuss the putative roles of conceptual knowledge, language, and sensorimotor experience already in the introduction to avoid ambiguity about the derived predictions and the chosen methodology. As it currently stands, I find it hard to discern how the presence of perceptual boundaries in LLMs could constitute evidence for affordance-based perception.

R6: The motivation of LLMs is to test the supramodality of this embodied discontinuity found in behavioral experiments: whether this discontinuity is accessible beyond the sensorimotor domain. To do this, we leveraged the recent advance in AI and tested whether the discretization observed in affordance perception is supramodally accessible to disembodied agents which lack access to sensorimotor input but only have access to the linguistic materials built upon discretized representations, such as large language models (LLM). The theoretical motivation and rationale regarding the LLM study are now included in the introduction and discussion:

In the introduction (p 2, ln 59) “…, and the body may serve as a metric that facilitates meaningful engagement with the environment by differentiating objects that are accessible for interactions from those not. Further, grounded cognition theory (see Barsalou, 2008 for a review) suggests that the outputs of such differentiation might transcend sensorimotor processes and integrate into supramodal concepts and language. From this perspective, we proposed two hypotheses...”

In the introduction (p 3, ln 70) “Notably, the affordance boundary varied in response to the imagined body sizes and showed supramodality. It could also be attained solely through language, as evidenced by the large language model (LLM), ChatGPT (OpenAI, 2023).”

For details in the discussion, please see R1.

(7) Along the same lines, the fMRI study also provides very limited evidence to support the authors' claims. The use of congruency effects as a way of probing affordance perception is not well motivated. What exactly can we infer from the fact a region may be more active when an object is paired with an activity that the object doesn't afford? The claim that "only the affordances of objects within the range of body size were represented in the brain" certainly seems far beyond the data.

R7: In our study, we followed the established fMRI research paradigm of employing the congruent effect as a measure of affordance processing (e.g., Kourtis et al., 2018).The choice of this paradigm has now been clarified in the revised manuscript (p 11, ln 244): “The congruency effect, derived from the contrast of Congruent versus Incongruent conditions, is a well-established measure of affordance processing (e.g., Kourtis et al., 2018).”

The statement that “only the affordances of objects within the range of body size were represented in the brain” is based on the observed interaction of congruency by object size. In the revised text, we have weakened this statement to better align with the direct implications of the interaction effect (p 1 ln 22): “A subsequent fMRI experiment revealed evidence of affordance processing exclusively for objects within the body size range, but not for those beyond. This suggests that only objects capable of being manipulated are the objects capable of offering affordance in the eyes of an organism.”

(8) Importantly (related to my comments under (2) above), the very small set of objects and affordances in this experiment heavily complicates any conclusions about object size being the crucial variable determining the occurrence of congruency effects.

R8: The objective of the fMRI study was to provide the neural basis for the affordance discontinuity found in behaviour experiments. In other words, the fMRI study is not an exploratory experiment, and therefore, the present object and action sets, which are based on the behaviour experiments, are sufficient.

(9) I would also suggest providing a more comprehensive illustration of the results (including the effects of CONGRUENCY, OBJECT SIZE, and their interaction at the whole-brain level).

R9: We agree and in revision, we have now included these analyses in the supplementary material (p 30, ln 711): “For the whole-brain analyses on the congruency effect, the object size effect, and their interaction, see Supplementary Fig. S4 and Table S2 to S5.” Please see Author response image 2 (New Supplementary Fig. S4) and Author response tables 3, 4 and 5 (New Supplementary Table S2 to S4) below.

**Author response table 3. sa4table3:** Cortical regions reaching significance in the contrasts of (A) objects within versus object beyond and (B) objects beyond versus objects within, whole-brain analysis (R = right hemisphere, L = left hemisphere; Z > 2.3, p = 0.05, cluster corrected).

Cluster	Region	Number ofvoxels in region	MNI coordinates	Peak Z value
x	y	z
Objects within >Objects beyond
1	L Cuneus	363	10	–90	20	4.24
R Cuneus	309
L Lingual Gyrus	176
R Lingual Gyrus	293
Objects beyond >Objects within
1	L Middle Occipital Gyrus	1831	–16	–92	-8	5.72
L Fusiform Gyrus	994
L Superior Parietal Lobule	584
L Lingual Gyrus	504
L Inferior Occipital Gyrus	304
L Superior Occipital Gyrus	211
L Parahippocampal Gyrus	210
L Precuneus	205
2	R Middle Occipital Gyrus	1358	20	–86	-2	5.4
R Lingual Gyrus	340
R Superior Parietal Lobule	328
R Superior Occipital Gyrus	320
R Inferior Occipital Gyrus	276
3	R Fusiform Gyrus	483	34	–38	–16	4.73
R Parahippocampal Gyrus	316

**Author response table 4. sa4table4:** Cortical regions reaching significance in contrasts of (A) congruent versus incongruent and (B) incongruent versus congruent, whole-brain analysis (R = right hemisphere, L = left hemisphere; Z > 2.3, p = 0.05, cluster corrected).

Cluster	Region	Number ofvoxels in region	MNI coordinates	Peak Z value
x	y	z
Congruent >Incongruent
1	L Inferior Parietal Lobule	322	–44	–50	64	3.47
2	R Superior Parietal Lobule	339	36	–66	52	3.31
R Inferior Parietal Lobule	165
Incongruent >Congruent
-	No significant cluster	-	-	-	-	-

**Author response table 5. sa4table5:** Review Table 5 (New Supplementary Table S4).Cortical regions showing significant interaction between object type and congruency, whole-brain analysis (OW = Objects within, OB = Objects beyond; R = right hemisphere, L = left hemisphere; Z > 2.3, p = 0.05, cluster corrected).

Cluster	Region	Number ofvoxels in region	MNI coordinates	Peak Z value
x	y	z
(OW_Congruent – OW_Incongruent) > (OB_Congruent – OB_Incongruent)
1	L Middle Occipital Gyrus	831	22	–94	10	4.25
R Middle Occipital Gyrus	187
L Fusiform Gyrus	113
R Fusiform Gyrus	376
L Inferior Occipital Gyrus	293
R Inferior Occipital Gyrus	276
L Lingual Gyrus	215
R Lingual Gyrus	345
L Superior Occipital Gyrus	123
2	R Supplementary Motor Area	383	14	14	60	3.39
3	R Superior Parietal Lobule	191	36	–62	56	3.18
R Inferior Parietal Lobule	114
4	R Insula	175	32	18	8	3.41
(OB_Congruent – OB_Incongruent) > (OW_Congruent – OW_Incongruent)
-	No significant cluster	-	-	-	-	-

**Author response image 2. sa4fig2:** Significant brain activations of different contrasts in the whole-brain level analysis. a, the effect of object type, positive values (warm color) indicated higher activation for objects within than objects beyond and negative values (cold color) indicated the opposite. b, the effect of congruency, positive values indicated higher activation in congruent than incongruent condition. c, the effect of interaction between object type and congruency, positive values indicated the larger congruency effect for objects within than beyond. d, the congruency effect for objects within. All contrasts were corrected with cluster-level correction at p < .05. The detailed cluster-level results for each contrast map can be found in Supplementary Table S2 to S5.

**Reviewer #3 (Recommendations For The Authors):**

a. (>a) Clarify all theoretical assumptions already within the introduction and specify how the predictions are tested (and how they could be falsified).

Ra: Please see R1.

b. (>b) Explain how the chosen experimental approach relates to the theoretical questions under investigation (e.g., it is not clear to me how affordance similarity ratings can inform inference about which part of the environment is perceived as more or less manipulable).

Rb: We thank the reviewer for the suggestion, and the theoretical motivation and rationale are now clarified. For details, please see R1.

c. (>c) Include a much larger set of objects and affordances in the behavioural experiments (that is more generalizable and also permits a more precise estimation of the boundary), and use a more rigorous methodology to justify a particular cluster solution.

Rc: Please see R2 for the limited variance of objects and actions, and R4 for more analyses on the boundary.

d. (>d) Clearly motivate what the use of LLMs can contribute to the study of affordance perception.

Rd: Please see R6.

(e) Clearly motivate why congruency effects are thought to index "affordance representation in the brain"Re: Please see R7.(e) Include a much larger set of objects and affordances in the fMRI study.

Re: Please see R7.

(f) Consider toning down the main conclusions based on the limitations outlined above.

Rf: We have toned down the main conclusions accordingly.

We are profoundly grateful for the insightful comments and suggestions provided by the three reviewers, which have greatly improved the quality of this manuscript. References

Barsalou, L. W. (1999). Perceptual symbol systems. Behavioral and Brain Sciences, 22(4), 637-660.

de Beeck, H. P. O., Torfs, K., & Wagemans, J. (2008). Perceived shape similarity among unfamiliar objects and the organization of the human object vision pathway. Journal of Neuroscience, 28(40), 10111-10123.

Borghi, A. M. (2005). Object concepts and action. Grounding cognition: The role of perception and action in memory, language, and thinking, 8-34.

Colling, L.J. (2021). ljcolling/go-bayesfactor: (Version v0.9.0).Zenodo. doi:10.5281/zenodo.4642331

Crawley, J. A. H., Mumby, H. S., Chapman, S. N., Lahdenperä, M., Mar, K. U., Htut, W., ... & Lummaa, V. (2017). Is bigger better? The relationship between size and reproduction in female Asian elephants. Journal of Evolutionary Biology, 30(10), 1836-1845.

Ellis, R., & Tucker, M. (2000). Micro‐affordance: The potentiation of components of action by seen objects. British Journal of Psychology, 91(4), 451-471.

Fan, L., Li, H., Zhuo, J., Zhang, Y., Wang, J., Chen, L., ... & Jiang, T. (2016). The human brainnetome atlas: a new brain atlas based on connectional architecture. Cerebral Cortex, 26(8), 3508-3526.

Fodor, J. A. (1975). The Language of Thought (Vol. 5). Harvard University Press.

Gibson, J. J. (1979). The ecological approach to visual perception: Classic edition.

Hertrich, I., Dietrich, S., & Ackermann, H. (2016). The role of the supplementary motor area for speech and language processing. Neuroscience & Biobehavioral Reviews, 68, 602-610.

Huang, T., Song, Y., & Liu, J. (2022). Real-world size of objects serves as an axis of object space. Communications Biology, 5(1), 1-12.

Kantak, S. S., Stinear, J. W., Buch, E. R., & Cohen, L. G. (2012). Rewiring the brain: potential role of the premotor cortex in motor control, learning, and recovery of function following brain injury. Neurorehabilitation and Neural Repair, 26(3), 282-292.

Kay, W., Carreira, J., Simonyan, K., Zhang, B., Hillier, C., Vijayanarasimhan, S., ... & Zisserman, A. (2017). The kinetics human action video dataset. arXiv preprint arXiv:1705.06950.

Konkle, T., & Oliva, A. (2011). Canonical visual size for real-world objects. Journal of Experimental Psychology: human perception and performance, 37(1), 23.

Kourtis, D., Vandemaele, P., & Vingerhoets, G. (2018). Concurrent cortical representations of function-and size-related object affordances: an fMRI study. Cognitive, Affective, & Behavioral Neuroscience, 18, 1221-1232.

Matić, K., de Beeck, H. O., & Bracci, S. (2020). It's not all about looks: The role of object shape in parietal representations of manual tools. Cortex, 133, 358-370.

McDannald, D. W., Mansour, M., Rydalch, G., & Bolton, D. A. (2018). Motor affordance for grasping a safety handle. Neuroscience Letters, 683, 131-137.

NCD Risk Factor Collaboration (NCD-RisC). (2016). A century of trends in adult human height. Elife, 5, e13410.

Peelen, M. V., & Caramazza, A. (2012). Conceptual object representations in human anterior temporal cortex. Journal of Neuroscience, 32(45), 15728-15736.

Rousseeuw, P. J. (1987). Silhouettes: a graphical aid to the interpretation and validation of cluster analysis. Journal of Computational and Applied Mathematics, 20, 53-65.

Sakreida, K., Effnert, I., Thill, S., Menz, M. M., Jirak, D., Eickhoff, C. R., ... & Binkofski, F. (2016). Affordance processing in segregated parieto-frontal dorsal stream sub-pathways. Neuroscience & Biobehavioral Reviews, 69, 89-112.

Van Gelder, T. (1998). The dynamical hypothesis in cognitive science. Behavioral and Brain Sciences, 21(5), 615-628.

Wagenmakers, E.-J., Wetzels, R., Borsboom, D. & van der Maas, H. L. J. Why psychologists must change the way they analyze their data: the case of psi: Comment on Bem (2011). Journal of Personality and Social Psychology, 100(3), 426–432.

Zhen, Z., Yang, Z., Huang, L., Kong, X. Z., Wang, X., Dang, X., ... & Liu, J. (2015). Quantifying interindividual variability and asymmetry of face-selective regions: a probabilistic functional atlas. NeuroImage, 113, 13-25.